# Learning Identifiable Balanced Prognostic Score for Treatment Effect Estimation Under Limited Overlap

## Abstract

Understanding individual-level treatment effects is a fundamental and crucial problem in causal inference. In this paper, our objective is to tackle the issue of limited overlap, where certain covariates only exist in a single treatment group. We demonstrate that, under weak conditions, it is possible to simultaneously recover identifiable balanced prognostic scores and balancing scores. By leveraging these scores, we relax the requirement of overlapping conditions in a latent space, enabling us to generalize beyond overlapped regions. This approach also allows us to handle *out-of-distribution* treatments with *no overlap*. Additionally, our approach is adaptable to various tasks, including both binary and structured treatment settings. Empirical results on different benchmarks demonstrate that our method achieves state-of-the-art performance.

## 1 Introduction

Treatment effect estimation plays a vital role in fields that require accurate decision making, such as medicine (Grzybowski et al., 2003), economics (Athey & Imbens, 2017), and education (Davies et al., 2018). The fundamental problem of causal inference (Holland, 1986) is that we can never observe the missing counterfactuals. Randomized control trials obviate these issues through randomization, but can be at times expensive (Sibbald & Roland, 1998) and impractical (Deaton & Cartwright, 2018). Therefore, deriving precise individual-level treatment effect from observational data is important and highly valuable.

The central challenge in causal inference from observational data is selection bias (Imbens & Rubin, 2015), where the distributions between treatment arms are different, *i.e.*, $p(t|x) \neq p(t)$. Previous studies have primarily focused on selection bias resulting from confounding variables, which are variables that causally affect both the treatment and outcome, and have relied on the unconfoundedness assumption (Rosenbaum & Rubin, 1983). However, instruments, which are covariates that causally affect only the treatment, can also introduce selection bias (Hassanpour & Greiner, 2019b). As we include more covariates that could potentially act as confounders or instruments, it becomes increasingly challenging to satisfy the requirement of overlapping support among treatments. Furthermore, in real-world scenarios, the treatment selection mechanism $p(t|x)$ that leads to selection bias can inherently lack overlap. For instance, a cautious doctor might not perform surgeries on elderly patients in all cases, making it difficult to generalize to surgical treatments for the elderly. As Pearl (2009) states, *"Whereas in traditional learning tasks we attempt to generalize from one set of instances to another, the causal modeling task is to generalize from behavior under one set of conditions to behavior under another set."* In the case of limited overlap, the causal model needs to generalize to previously unadministered treatments, which can even be completely different, and this challenge frequently arises in structured settings (Ramsundar et al., 2019).

Previous approaches aimed at mitigating selection bias often assume unconfoundedness and overlook the issue of limited overlap. Reweighting-based methods (Farrell, 2015; Gretton et al., 2009) typically rely on the presence of common support between the treatment and control groups to adjust for distribution mismatch. Subsequently, there has been an increasing interest in *balanced representation learning* since Johansson et al. (2016). However, most of these methods primarily tackle selection bias and do not explicitly consider the problem of limited overlap. Wu & Fukumizu

(2021) stands as a pioneering work that considers limited overlap in the *within-sample* setting by learning an *entangled prognostic score* (Hansen, 2008).

To effectively address selection bias, including the potential challenge of limited overlap, we employ a latent identifiable generative model (Khemakhem et al., 2020) that simulateously learns identifiable *balancing score* and *balanced prognostic score* by disentangling $X$. Identifiable balancing score is naturally obtained by concatenating identifiable instruments and confounders, while identifiable balanced prognostic score is obtained by concatenating identifiable confounders and adjustments. Intuitively, modeling identifiable balancing score helps us identify the root cause of selection bias, while modeling identifable balanced prognostic score enables us to directly estimate the outcome by leveraging the learned identifiable disentangled representation that are direct causes of the outcome $Y$.

Our contributions can be summarized as follows: i) We demonstrate that, under weak conditions, it is possible to simultaneously recover the identifiable balanced prognostic score and balancing score. Furthermore, we provide theoretical results on how a balanced prognostic score effectively handle the limited overlap problem. ii) We introduce a practical and generalized disentanglement method called Disentangled Identifiable vaRiational autoEncoder (DIRE). This method is designed to model the data generation process with identifiability guarantee. iii) We apply our method to both binary and structured treatment settings. Notably, we demonstrate how an identifiable balanced prognostic score can generalize to *out-of-distribution* treatments with *zero overlap*, showcasing its robustness. iv) Through comprehensive experiments, we demonstrate that our method outperforms other state-of-the-art models in their respective settings. This superiority is evident in both the widely-used de facto binary treatment benchmark and various limited overlapping synthetic datasets. Synthetic datasets, along with code, will be made publicly available upon publication.

## 2 RELATED WORK

There are two main approaches to addressing selection bias. One approach involves sample reweighting to align different distributions. A common method within this approach is to use propensity scores for inverse weighting of samples (Rosenbaum & Rubin, 1983; Austin, 2011; Allan et al., 2020; Freedman & Berk, 2008). However, weighting based on propensity scores can be unstable and lead to high variance (Swaminathan & Joachims, 2015). To address this issue, researchers have proposed more stable weighting methods. For instance, Gretton et al. (2009) reweights samples to achieve distribution matching in a high dimensional feature space, while Zubizarreta (2015) learns weight that minimizes variance and balances distributions simultaneously. Athey et al. (2018) combines sample reweighting and regression adjustments through approximate residual balancing, offering the benefits of both approaches.

Ever since Johansson et al. (2016), there has been an growing interest in mitigating selection bias via minimizing distribution discrepancy (Mansour et al., 2009) of learned representations (Bengio et al., 2013). Shalit et al. (2017) improve upon Johansson et al. (2016)'s work by learning treatment-specific function on top of a prognostic score (Hansen, 2008), so that the treatment bit does not get lost in the distribution alignemnt stage. Hassanpour & Greiner (2019b) proposes learning disentangled representations to clearly identify factors that contribute to either the treatment $T$, the outcome $Y$, or both, in order to to better account for selection bias and achieve improved result. Wu & Fukumizu (2021) provides identification guarantee in **within-sample** setting, learning a prognostic score whose dimension is not higher than that of the outcome $Y$. In our work, we aim to learn disentangled representation with causal generative process that adheres to the *independent causal mechanism* (Schölkopf et al., 2021). Disentangled representation is preferred because, unlike entangled representations, it allows for sparse or localized changes in the causal factors when the distribution undergoes interventions (Schölkopf et al., 2021), making our model more robust to such changes.

Several approaches have been proposed to address the limited overlapping problem. Crump et al. (2009) suggests using optimal sub-samples to estimate the average treatment effect. Grzybowski et al. (2003) excludes patients whose propensity scores cannot be matched. Jesson et al. (2020) focuses on identifying the limited overlapping regions without providing estimations. Oberst et al. (2020) provides an interpretable characterization of the distributional overlap between treatment groups.

## 3 PRELIMINARIES

Our objective is to estimate $\mathbb{E}[Y(t)|X]$ for all $x \in \mathcal{X}$ and $t \in \mathcal{T}$, where $x_i, t_i, y_i$ represents our dataset with $x_i$ as the observed covariates, $t_i$ as the administered treatment, and $y_i$ as the corresponding outcome. This estimation allows us to accurately assess $\mathbb{E}[Y(t_i) - Y(t_j)|X]$ for all $t_i, t_j \in \mathcal{T}$ and $x \in \mathcal{X}$. Here, $Y(t)$ refers to the potential outcome, representing the hidden value that would have been observed if $T = t$ was administered. By applying the backdoor criterion (Pearl, 2009) to the causal graph depicted in Figure-1, we can identify the individual-level treatment effect once we recover $Z_2$ and $Z_1$.

We adopt the generalized definition of overlapping condition from Wu & Fukumizu (2021):

**Definition 1** $V$ is overlapping if $P(T|V = v) > 0$ for any $t \in \mathcal{T}, v \in \mathcal{V}$. If the condition is violated at some value $v$, then $v$ is non-overlapping and $\mathcal{V}$ is limited-overlapping.

As such, to accurately estimate the treatment effect, it is preferable to obtain a lower-dimensional representation (Bengio et al., 2013) that exhibits overlap, even if the original covariate space is limited overlapping.

We adapt Wu & Fukumizu (2021)'s definition of prognostic score (Hansen, 2008) to accommodate for multiple treatments:

**Definition 2** A prognostic score (PGS) is $\{p(X, t)\}_{t \in \mathcal{T}}$, such that $Y(t) \perp\!\!\!\perp X \mid p(X, t)$, where $p(X, t)$ is a function defined on $\mathcal{X} \times \mathcal{T}$. A PGS is called Balanced Prognostic Score (bPGS) if $p(x, t_i) = p(x, t_j)$ for all $t_i, t_j \in T$

Since the prognostic score serves as a sufficient statistic for the outcome $Y$, it is only necessary to fulfill the overlapping condition over prognostic scores, rather than over the covariates themselves. Intuitively, requiring overlap over all covariates may be overly strict, as some of them may be generated by underlying instrumental latent factors and therefore irrelevant for estimating the outcome. We will demonstrate this in a mathematically rigorous manner later on.

## 4 METHODOLOGY

In this section, we offer a comprehensive introduction to our method. We begin by presenting the assumptions of the data generating process in Sec 4.1. Following that, in Sec 4.2, we demonstrate how a balanced prognostic score tackles the issue of limited overlap. Finally, in Sec 4.3, we present our model architecture that offers identifiability guarantee and provide a concise overview of its implementation.

### 4.1 DATA GENERATING PROCESS AND SETUP

We assume that the Data Generating Process (DGP) follows the causal graph presented in Fig. 1(a). In this graph, the covariate $X$ is generated from three latent variables: $Z_1$ (adjustment variable), $Z_2$ (confounder variable), and $Z_3$ (instrumental variable). The outcome $Y$ is generated by $Z_1$ and $Z_2$, while the treatment $T$ is generated by $Z_2$ and $Z_3$. Mathematically, the DGP assumptions can be formulated as follows:

**Assumption 4.1** (DGP for covariates) The covariates are generated from underlying ground-truth latent code $Z_1$ (adjustment variable), $Z_2$ (confounder variable), $Z_3$ (instrumental variable), where

$$X = \tilde{K}(\tilde{Z}_1, \tilde{Z}_2, \tilde{Z}_3) = \tilde{K}_1(\tilde{Z}_1) \oplus \tilde{K}_2(\tilde{Z}_2) \oplus \tilde{K}_3(\tilde{Z}_3) \oplus \tilde{K}_4(\tilde{Z}_1, \tilde{Z}_2) \oplus \tilde{K}_5(\tilde{Z}_1, \tilde{Z}_3)$$
$$\oplus \tilde{K}_6(\tilde{Z}_2, \tilde{Z}_3) \oplus \tilde{K}_7(\tilde{Z}_1, \tilde{Z}_2, \tilde{Z}_3) + e_1. \tag{1}$$

In DIRE, we intend to model $\tilde{Z}_1$, $\tilde{Z}_2$ and $\tilde{Z}_3$, and the data generating process $\tilde{K}$:

$$X = K(Z_1, Z_2, Z_3) = K_1(Z_1) \oplus K_2(Z_2) \oplus K_3(Z_3) \oplus K_4(Z_1, Z_2) \oplus K_5(Z_1, Z_3)$$
$$\oplus K_6(Z_2, Z_3) \oplus K_7(Z_1, Z_2, Z_3) + \epsilon_1. \tag{2}$$

where $\oplus$ denotes dimension concatenation. The random variables and mappings denoted by ˜ represent the ground-truth latent factors and mapping, while those without the symbol represent the learned parameters. Consistent with the works of Wu & Fukumizu (2021) and Khemakhem et al. (2020), we assume $K_1$-$K_7$ to be injective.

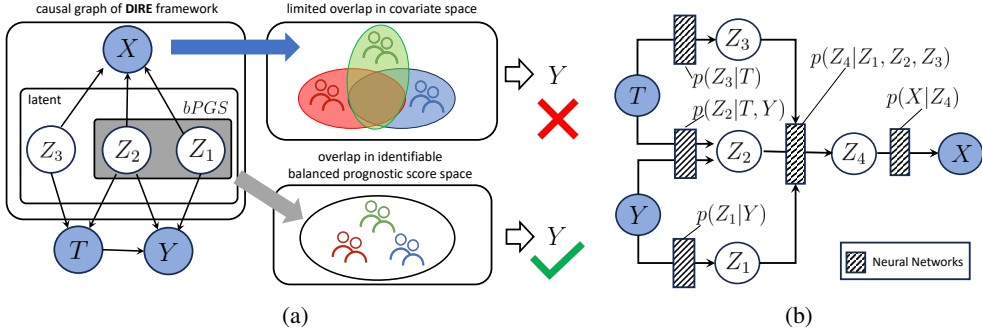

Figure 1: (a) Extracting identifiable bPGS from latent factors via reconstructing covariates. (b) DIRE decoder model architecture.

**Assumption 4.2** (DGP for Y) The outcome is generated from underlying ground-truth latent code $\tilde{Z}_1, \tilde{Z}_2$:

$$Y = \tilde{J}(\tilde{Z}_1, \tilde{Z}_2, T) = \tilde{j}_t(\tilde{Z}_1, \tilde{Z}_2) + e_2 = \tilde{j}_t \circ p + e_2, \tag{3}$$

where the second equality is obtained through application of do-calculus (Pearl, 2009) in Fig. 1, and has been shown in Zhang et al. (2021). This is essentially a relaxation of assumption **(G1')** in Wu & Fukumizu (2021) without assuming $j_t$ being injective.

Similarly, we have:

$$Y = J(Z_1, Z_2, T) = j_t(Z_1, Z_2) + \epsilon_2. \tag{4}$$

**Assumption 4.3** (DGP for T) The treatment is generated from underlying ground-truth latent code $Z_2, Z_3$, where

$$T = \tilde{M}(\tilde{Z}_2, \tilde{Z}_3) + e_3, \tag{5}$$

and

$$T = M(Z_2, Z_3) + \epsilon_3. \tag{6}$$

This assumption is just a mathematical formulation of directed edges $(Z_3, T)$ and $(Z_2, T)$ in Fig. 1.

Finally, inspired by Kaddour et al. (2021), we make the following assumption:

**Assumption 4.4** (Product effect for prognostic score) $\forall p \in \{p(x, t)\}$, $p$ can be factorized as:

$$p = (g_1(X)^T h_1(T), g_2(X)^T h_2(T), \ldots, g_n(X)^T h_n(T)) + \epsilon, \tag{7}$$

$$= (g_1(X), \cdots, g_n(X)) \begin{bmatrix} h_1(T) & \cdots & 0 \\ \vdots & \ddots & \vdots \\ 0 & \cdots & h_n(T) \end{bmatrix} + \epsilon, \tag{8}$$

where there exists Reproducing Kernel Hilbert Space $H_{\mathcal{X}}$ and $H_{\mathcal{T}}$ such that $g_i(X) \in H_{\mathcal{X}}$ and $h_i(T) \in H_{\mathcal{T}}$ for $1 \leq i \leq n$. This assumption is considered mild, as highlighted in Kaddour et al. (2021). Subsequently, we will explore the universality of this assumption and demonstrate the relationship between prognostic score (PGS) and balanced prognostic score (bPGS) under this assumption.

## 4.2 Identifications Under Limited-overlapping Covariate

Limited overlap is a common occurrence in treatment effect estimation scenarios that involve high-dimensional covariates and multiple potential treatments. In this subsection, we initially illustrate how the requirement for overlap can be relaxed within a latent space. Furthermore, we demonstrate how the presence of an identifiable balanced prognostic score (bPGS) enables us to extend our generalization beyond regions of overlap.

We first establish the generality of **Assumption 4.4**, and how we can derive a balanced prognostic score using a prognostic score.

**Proposition 1** (Universality of product effect formalization for prognostic score) Let $\mathcal{H}_{\mathcal{X} \times \mathcal{T}}$ be the given Reproducing Kernel Hilbert Space. For any $\epsilon > 0$ and any $f \in \mathcal{H}^n$, there is a $d \in \mathbb{N}$ such that there exist $2n$ d-dimensional function $g_i : \mathcal{X} \rightarrow \mathbb{R}^d$ and $h_i : \mathcal{T} \rightarrow \mathbb{R}^d$ such that $\|f - (g_i^T h_i, \ldots g_n^T h_n)\|_{L_2(P_{\mathcal{X} \times \mathcal{T}})} \leq \epsilon$.

Thus, when provided with a prognostic score (PGS) $p_t \in p(x, t)$, we can always derive a balanced prognostic score (bPGS) $(g_1(X), \ldots, g_n(X))$. Referring to Fig. 1, we can interpret the learning of the bPGS as the inverse mapping of the generative process for the covariates $\mathcal{X}$. In other words, our model is inclined to acquire a more general bPGS, rather than just a PGS, which can be utilized for the downstream CATE task.

In the following theorem, we show how learning bPGS enable us to relax the overlapping condition, and how bPGS enable us to generalize beyond non-overlapping regions, which frequently occurs in mutiple and structured treatment setting.

**Theorem 1** Suppose **Assumption 4.1** - **Assumption 4.4** hold. Furthermore, $\tilde{K}_i$ and $K_i$ are injective for all $i$. Then if $\mathbb{E}_{p_\theta}[X|Z_1, Z_2, Z_3] = \mathbb{E}[X|\tilde{Z}_1, \tilde{Z}_2, \tilde{Z}_3]$, we have:

1. (Recovery of latent code) If either
   1) $\tilde{K}_1$, $\tilde{K}_2$ and $\tilde{K}_3$ are not empty mapping, or
   2) at least two of $\tilde{K}_4$-$\tilde{K}_7$ are non-empty mappings, $I(\Delta_T \tilde{Z}_1; T) = 0$, $I(\Delta_Y \tilde{Z}_3; Y|T) = 0$ for some injective $\Delta_T$ and $\Delta_Y$, $I(Z_2; T) \neq 0$ and $I(Z_2; Y) \neq 0$,
   then $Z_1 = \Delta_1 \circ \tilde{Z}_1$, $Z_2 = \Delta_2 \circ \tilde{Z}_2$, $Z_3 = \Delta_3 \circ \tilde{Z}_3$ for some injective mapping $\Delta_1, \Delta_2, \Delta_3$.

2. (Recovery of bPGS via subset of covariates) $Z = Z_1 \oplus Z_2 = v \circ p$ for some injective mappint $v$. Moreover, the overlapping condition can be relaxed onto $X' \subseteq X$ where where $\mathcal{X}' := \{x \in \mathcal{X}|k_4^{*-1}(x) \text{ is overlapping}\} \cup \{x \in \mathcal{X}|k_1^{*-1}(x) \text{ and } k_2^{*-1}(x) \text{ is overlapping}\} \cup \{x \in \mathcal{X}|k_7^{*-1}(x) \text{ is overlapping}\}$.

3. (OOD generalization on non-overlapping regions) Suppose $\tilde{f}_t(x) = \mathbb{E}[Y|X, T] = E_{p_\theta}[Y|X, T] = f_t(x)$ for all observed $(x, t) \in \mathcal{X} \times \mathcal{T}$. Suppose $\exists t' \in \mathcal{T}$ s.t. $j_t'$ and $\tilde{j}_t'$ are injective. Suppose there exist a RKHS $\mathcal{H}_\mathcal{P}$ on the bPGS space, also $\tilde{j}_t^* \in \mathcal{H}_\mathcal{P}$ and $j_t^* \circ \Delta \in \mathcal{H}_\mathcal{P}$ for all $t^* \in \mathcal{T}$ where $\Delta := j_t'^{-1} \circ \tilde{j}_t'$. Then we have $||j_t \circ \Delta - \tilde{j}_t|| < \epsilon \Rightarrow |\tilde{f}_t(x) - f_t(x)| < \epsilon * C$ for some constant $C$ for all $t \in \mathcal{T}$.

According to **Theorem 1**, the requirement for overlap can be relaxed to the variables $Z_1$ and $Z_2$. Furthermore, the acquisition of a balanced prognostic score (bPGS) allows for generalization to limited overlapping regions, as long as $j_t$ can be recovered. In our structured treatment setting, we empirically demonstrate that our recovered bPGS enables generalization even to *out-of-distribution* $j_t$ values with *zero overlap*, highlighting the advantages of learning an identifiable balanced prognostic score.

## 4.3 MODEL ARCHITECTURE AND IMPLEMENTATION

To recover the underlying instrumental variables, confounding variables, and adjustment variables, we propose a method named Disentangled Identifiable vaRiational autoEncoder (DIRE) to reconstruct the covariates. In DIRE, we leverage treatment and outcome information as auxiliary supervision signals to guide the learning process and recover the identifiable latent factors. This process is illustrated in Fig. 1(b).

Put more formally, Let $\theta = (f, g, T, \lambda)$ be parameters of the following generative model:

$$p_\theta(x, z_1, z_2, z_3, z_4|t, y) = p_{T,\lambda}(z_1|y)p_{T,\lambda}(z_2|t, y)p_{T,\lambda}(z_3|t)p_g(z_4|z_1, z_2, z_3)p_f(x|z_4), \quad (9)$$

where we assume:

$$p_\epsilon(x - f \circ g(z_1, z_2, z_3)) = p_f(x|z_4)p_g(z_1, z_2, z_3), \quad (10)$$

$$p_{T,\lambda}(z_1, z_2, z_3|t, y) = p_{T,\lambda}(z_1|y)p_{T,\lambda}(z_2|t, y)p_{T,\lambda}(z_3|t), \quad (11)$$

where in Eq.10 $f$ and $g$ are injective, and in Eq.11 we are requiring the generative process to be consistent with our causal model. The graphical model of decoder is shown in Fig. 1

The corresponding inference model factorizes as:

$$q_\phi(z_1, z_2, z_3, z_4|x, t, y) = q_\phi(z_4|x)q_\phi(z_1|x, t)q_\phi(z_2|z_4)q_\phi(z_3|z_4, y). \tag{12}$$

Incoporating the ELBO decomposition trick (Chen et al., 2018) to better isolate the irrelevant factors from $\mathcal{X}$ from the latent factors of interest, we have

**Theorem 2** The ELBO of DIRE is

$\mathbb{E}_{p(x)p(t|x)p(y|t,x)}[p_\theta(x|t, y)] \geq$

$\mathbb{E}_{p(x,t,y)q_\phi(z_4|x)}[\log p_{\theta(x|z_4)}] + \mathbb{E}_{q_\phi(z_1,z_2,z_3,z_4,x,t,y)}[\log p_\theta(z_4|z_1, z_2, z_3) - \log q_\phi(z_4|x)]$

$+ \sum_{i=1}^{3} \mathbb{E}_{p(x,t,y)}\mathbb{E}_{q_\phi(z_4|x)}[-KL(q_\phi(z_i|pa_\phi(z_i))||q_\phi(z_i)) - \sum_j KL(q_\phi(z_{ij})||p_\theta(z_{ij}|pa(z_{ij})))$

$- KL(q_\phi(z_i)|| \prod_j q_\phi(z_{ij}))], \tag{13}$

where $pa(z)$ denote the parent nodes of $z$ in Fig 1.

Given auxiliary information $T, Y$, the learned latent factors are identifiable.

**Proposition 2** Assume the following hold:

- $f$ and $g$ are injective in Eq.10.

- Let $\psi_\epsilon$ be the characteristic function of $p_\epsilon$. $\{x \in \mathbb{X}|\psi_\epsilon(x) = 0\}$ has measure zero.

- Suppose $z_1 \in \mathbb{R}^a$, $z_2 \in \mathbb{R}^b$, and $z_3 \in \mathbb{R}^c$, $a + b + c = n$, then $\lambda(t, y) = \lambda_1(y) \oplus \lambda_2(t, y) \oplus \lambda_3(t)$, where $\lambda_1(y) \in \mathbb{R}^{2a}$, $\lambda_2(t, y) \in \mathbb{R}^{2b}$, $\lambda_3(t) \in \mathbb{R}^{2c}$ are parameters of gaussian distribution.

- There exists $2n + 1$ points $(t_0, y_0) \ldots (t_{2n+1}, y_{2n+1})$ such that the matrix $L = [(\lambda_1(y_1) - \lambda_1(y_0)) \oplus (\lambda_2(t_1, y_1) - \lambda_2(t_0, y_0)) \oplus (\lambda_3(t_1) - \lambda_3(t_0)), \ldots, (\lambda_1(y_{2n+1}) - \lambda_1(y_0)) \oplus (\lambda_2(t_{2n+1}, y_{2n+1}) - \lambda_2(t_0, y_0)) \oplus (\lambda_3(t_{2n+1}) - \lambda_3(t_0)))] = [(\lambda(t_1, y_1) - \lambda(t_0, y_0)), \ldots, (\lambda(t_{2n+1}, y_{2n+1}) - \lambda(t_0, y_0))]$ is invertible, i.e., $\lambda = \lambda_1 \oplus \lambda_2 \oplus \lambda_3$ where $\lambda_1$ is independent of $t$, and $\lambda_3$ is independent of $y$.

- The sufficient statistics are differentiable almost everywhere.

- Let $k = f \circ g$, then $k(z_1, z_2, z_3) = k_1(z_1) \oplus k_2(z_2) \oplus k_3(z_3) \oplus k_4(z_1, z_2) \oplus k_5(z_1, z_3) \oplus k_6(z_2, z_3) \oplus k_7(z_1, z_2, z_3)$ satisfies $Range(k_i) \cap Range(k_j) = \emptyset$.

then if $p_\theta(x|t, y) = p'_\theta(x|t, y)$ we have

$$k^{-1}(x) = diag(a)k'^{-1}(x) + b. \tag{14}$$

Hence, agreement on observational distribution, in our case the covariates $\mathcal{X}$, implies that the underlying generating model parameter is uniquely determined. Moreover, as indicated in (A) such identification can be done up to translation and scaling.

The derivation of the derived ELBO in **Theorem 1** enables us to learn identifiable latent representations for adjustments, confounders, and instruments. We add two estimators on top of the balanced prognostic score and balancing score. Estimating the selected treatment using the balancing score allows us to more accurately identify the root cause of selection bias. Furthermore, estimating the outcome using the balanced prognostic score enables us to obtain more robust outcome estimations across different treatments.

The overall loss is derived as:

$$\mathcal{L} = \mathcal{L}_{prognostic\ score} + \mathcal{L}_{ELBO} + \mathcal{L}_{balancing\ score}. \tag{15}$$

And the loss for the ELBO is:

$\mathcal{L}_{ELBO} =$

$$\mathbb{E}_{p(x,t,y)q_\phi(z_4|x)}[\log p_\theta(x|z_4)] - \mathbb{E}_{q_\phi(z_1,z_2,z_3,x,t,y)}[\alpha_4(\log q_\phi(z_4|x) - \log q_\phi(z_4))$$

$$+ \beta_4(\log q_\phi(z_4) - \log q_\phi(\prod_j z_{4j})) + \gamma_4(\log q_\phi(\prod_j z_{4j}) - \log p_\theta(z_4|z_1, z_2, z_3))]$$

$$+ \sum_{i=1}^{3} \alpha_i \mathbb{E}_{p(x,t,y)} \mathbb{E}_{q_\phi(z_4|x)}[-KL(q_\phi(z_i|pa_\phi(z_i))||q_\phi(z_i)) - \beta_i \sum_j KL(q_\phi(z_{ij})||p_\theta(z_{ij}|pa(z_{ij})))$$

$$- \gamma_i KL(q_\phi(z_i)||\prod_j q_\phi(z_{ij}))]. \tag{16}$$

in which we introduced the ELBO decomposition trick (Chen et al., 2018) to learn better disentangled representations.

$\mathcal{L}_{prognostic\ score}$ is the loss of the outcome predictor, where we can use the loss function of any downstream treatment effect estimators such as (Shalit et al., 2017; Hassanpour & Greiner, 2019a; Künzel et al., 2019; Yao et al., 2018), and $\mathcal{L}_{balancing\ score}$ is the loss of the treatment predictor, where we predict the treatment using the identifiable balancing score.

## 5 EXPERIMENTS

Our experiments aim to answer the following questions: Q1: Can our method effectively handle the limited overlap problem? Q2: Is our method robust when faced with varying degrees of limited overlap? Q3: Can our method successfully address the limited overlap problem within the structured treatment setting? Q4: How does our method perform in scenarios with zero overlap? To evaluate our approach, we conduct experiments on synthetic and semi-synthetic datasets, considering both within-sample and out-sample settings.

### 5.1 EXPERIMENTAL SETUP

**Dataset.** We conducted experiments on three datasets, and the detailed information can be found in the Appendix. First, IHDP, a de facto semi-synthetic benchmark compiled by Hill (2011) to study the treatment effect of home visit on future cognitive test scores. We follow the same setting as Johansson et al. (2016); Shalit et al. (2017); Louizos et al. (2017), averaging over 1000 replications of simulated outcomes with a 63/27/10 train/validation/test split. Second, we synthesized a more challenging synthetic dataset to assess the performance of our method under different degrees of limited overlap. Third, drawing inspiration from Kaddour et al. (2021), we designed a structured treatment dataset using scaffold split (Ramsundar et al., 2019). This dataset required us to perform zero-shot/zero-overlap treatment effect estimation on *out-of-distribution* treatments. For further details regarding the synthetic datasets, please refer to the Appendix.

**Baselines.** We choose BLR, BNN (Johansson et al., 2016), BART (Chipman & McCulloch, 2016; Chipman et al., 2010), RF (Breiman, 2001), CF (Wager & Athey, 2018), CEVAE (Louizos et al., 2017), GANITE (Yoon et al., 2018), $\beta$-intact-VAE (Wu & Fukumizu, 2021), DR-CFR (Hassanpour & Greiner, 2019b), SIN (Kaddour et al., 2021) as baselines. In particular, we included $\beta$-intact-VAE as a comparable baseline that primarily addresses limited overlap. SIN was chosen due to its ability to handle structured treatment settings. We also selected DR-CFR, a disentanglement learning method, to compare its performance against our proposed DIRE in the limited overlap setting.

### 5.2 RESULTS ON IHDP (Q1)

We adopt two metrics to evaluate the methods. Individual-based evaluation metric, $PEHE = \sqrt{\sum_{i=1}^{N}((y_{1i} - y_{0i}) - (\tau_{1i} - \tau_{0i}))^2}$ and population-based metric, $\epsilon_{ATE} = |\sum_{i=1}^{N}(\tau_{1i} - \tau_{0i}) - \sum_{i=1}^{N}(y_{1i} - y_{0i})|$. Results[1] are depicted in Tab. 2, where the best results for each metric is bolded, and the runner-ups are underlined.

---

[1]Results are taken directly from Shalit et al. (2017); Louizos et al. (2017); Yoon et al. (2018); Wu & Fukumizu (2021).

Table 1: IHDP Resuts.

| Method | within-sample | | out-sample | |
|---|---|---|---|---|
| | *PEHE* | $\epsilon_{ATE}$ | *PEHE* | $\epsilon_{ATE}$ |
| OLS-1 | $5.8 \pm .3$ | $.73 \pm .04$ | $5.8 \pm .3$ | $.94 \pm .06$ |
| OLS-2 | $2.4 \pm .1$ | $.14 \pm .01$ | $2.5 \pm .1$ | $.31 \pm .02$ |
| BLR | $5.8 \pm .3$ | $.72 \pm .04$ | $5.8 \pm .3$ | $.93 \pm .05$ |
| k-NN | $2.1 \pm .1$ | $.14 \pm .01$ | $4.1 \pm .2$ | $.79 \pm .05$ |
| BART | $2.1 \pm .1$ | $.23 \pm .01$ | $2.3 \pm .1$ | $.34 \pm .02$ |
| RF | $4.2 \pm .2$ | $.73 \pm .05$ | $6.6 \pm .3$ | $.96 \pm .06$ |
| CF | $3.8 \pm .2$ | $.18 \pm .01$ | $3.8 \pm .2$ | $.40 \pm .03$ |
| BNN | $2.2 \pm .1$ | $.37 \pm .03$ | $2.1 \pm .1$ | $.42 \pm .03$ |
| CFR-WASS | $.71 \pm .0$ | $.25 \pm .01$ | $.76 \pm .0$ | $.27 \pm .01$ |
| CEVAE | $2.7 \pm .1$ | $.34 \pm .01$ | $2.6 \pm .1$ | $.46 \pm .02$ |
| GANITE | $1.9 \pm .4$ | $.43 \pm .05$ | $2.4 \pm .4$ | $.49 \pm .05$ |
| Beta-Intact-VAE | $0.709 \pm .024$ | $.180 \pm .007$ | $0.946 \pm .048$ | $.211 \pm .011$ |
| DIRE | $\mathbf{0.475 \pm 0.006}$ | $\mathbf{0.130 \pm 0.003}$ | $\mathbf{0.520 \pm 0.011}$ | $\mathbf{0.141 \pm 0.003}$ |

As shown in Table 2, DIRE consistently outperforms all other baseline methods across all evaluation metrics. Notably, even though Wu & Fukumizu (2021) primarily focuses on the post-treatment setting, DIRE achieves a significant improvement over $\beta$-Intact-VAE. Furthermore, since DIRE also generalizes its identification capability to the out-sample setting, we have achieved state-of-the-art (SOTA) results in the out-sample scenario as well.

## 5.3 RESULTS ON SYNTHETIC DATASET (Q2)

To assess the effectiveness of our method across different degrees of limited overlap, we conducted experiments using five non-overlapping levels denoted as $\omega$, where a higher value of $\omega$ indicates a more severe non-overlapping scenario. For each non-overlapping level, we examined 27 configurations by varying the dimensions of the latent variable, specifically $dim\ v \in \{4, 8, 10\}$.

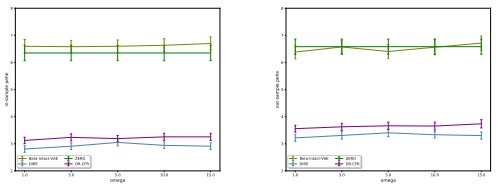

Our data generation process differs from that of Wu et al. (2021) in that we also consider $Z_3$ as a source of selection bias. This additional factor makes it more challenging to derive a low-dimensional balanced prognostic score from the covariates.

Figure 2: Synthetic Dataset Result.(a) In sample (b) Out sample.

To ensure fair comparison, we conduct hyperparameter search using Li et al. (2020) on a hold-out validation dataset and select the best hyperparameters over 30 runs. The results, depicted in Figure 2, include both in-sample (Figure 2(a)) and out-sample (Figure 2(b)) evaluations.

We observed that even in the in-sample scenario, $\beta$-Intact-VAE struggles to generate a balanced prognostic score in the presence of instruments, where the overlapping condition is not necessary. The performance of DR-CFR diminishes as the limited overlapping level becomes more severe, as evident from Figure 2 when $\omega$ is set to 10 or 15. In contrast, DIRE exhibits robustness across all limited overlapping levels, with its performance remaining unaffected or even improving in more severe cases. This highlights the efficacy of learning a balanced prognostic score and a balancing score simultaneously in DIRE.

## 5.4 RESULTS ON STRUCTURED TREATMENTS DATASET (Q3&Q4)

The structured treatment setting presents additional challenges due to the involvement of multiple treatments, where even slight variations in the treatment structure result in a different treatment. As

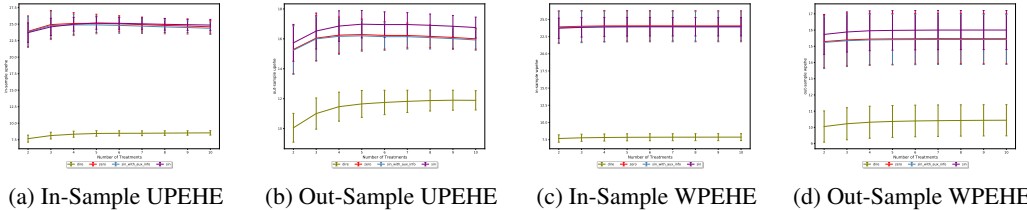

| (a) In-Sample UPEHE | (b) Out-Sample UPEHE | (c) In-Sample WPEHE | (d) Out-Sample WPEHE |

Figure 3: Structured dataset result averaged over 25 random seeds.

such, we investigate the *out-of-distribution* treatment setting to see if our learned balanced prognostic score enables us to generalize under the out-of-distribution zero-shot setting. Given that $\beta$-intact-VAE (Wu & Fukumizu, 2021) cannot handle the structured treatment problem, We mainly compare with SIN (Kaddour et al., 2021) whose $g(X)$ representation naturally serves as a balanced prognostic score as well.

We use the evaluation metric proposed by Kaddour et al. (2021), where $\epsilon_{\text{UPEHE(WPEHE)}} = \int_{\mathcal{X}} (\hat{\tau}(t', t, x) - \tau(t', t, x))^2 p(t|x)p(t'|x)p(x)dx$. *PEHE@K* is computed over the top $K$ treatments ranked by propensities with $\binom{K}{2}$ combinations. To ensure fair comparison, we conduct hyperparameter search using Li et al. (2020) on a hold-out validation dataset and select the best hyperparameters over 100 runs. For more detail refer to the appendix. The results are shown in Tab. 3, where the best results for each metric is bolded, and the runner-ups are underlined.

Table 2: CATE Estimation Error measured at PEHE@10, averaged over 25 random seeds.

| Method | Weighted PEHE | | Unweighted PEHE | |
|---|---|---|---|---|
| | *Within-Sample* | *Out-Sample* | *Within-Sample* | *Out-Sample* |
| ZERO | $24.05 \pm 2.20$ | $15.47 \pm 1.54$ | $24.60 \pm 0.97$ | $16.00 \pm 0.69$ |
| SIN | $23.93 \pm 1.33$ | $16.00 \pm 1.20$ | $24.86 \pm 0.85$ | $16.76 \pm 0.70$ |
| SIN-With-Aux-Info | $23.94 \pm 2.19$ | $15.42 \pm 1.53$ | $24.38 \pm 0.95$ | $15.93 \pm 0.69$ |
| DIRE | $\mathbf{7.87 \pm 0.50}$ | $\mathbf{10.44 \pm 0.96}$ | $\mathbf{8.54 \pm 0.33}$ | $\mathbf{11.89 \pm 0.65}$ |

SIN does not effectively utilize the auxiliary information and performs worse than zero. Even when provided with auxiliary information $T$ (a vector of molecular properties used as the treatment), SIN still struggles to learn a stable balanced prognostic score (bPGS), with its performance being similar to zero.

In contrast, DIRE successfully identifies the confounding factors even when faced with out-of-distribution treatment $j_t$ in the zero-overlapping scenario, as outlined in Assumption 4.2. This demonstrates that only DIRE effectively learns a balanced prognostic score, while the other methods fall short in this regard.

## 6 CONCLUSION

This paper addresses the challenge of limited overlap in treatment effect estimation by proposing a method that allows for the identification of latent adjustments, confounders, and instruments. By leveraging these latent factors, we can relax the requirement of overlapping conditions and extend our estimation to non-overlapping regions. Moreover, our method enables generalization to out-of-distribution treatments with zero overlap. The experimental results demonstrate the superiority of our proposed method across various benchmarks, highlighting its effectiveness and versatility.

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

## A    THEORETICAL RESULTS

**Theorem 2**    The ELBO of DIRE is

$\mathbb{E}_{p(x)p(t|x)p(y|t,x)}[p_\theta(x|t,y)] \geq$

$\mathbb{E}_{p(x,t,y)q_\phi(z_4|x)}[\log p_{\theta(x|z_4)}] + \mathbb{E}_{q_\phi(z_1,z_2,z_3,z_4,x,t,y)}[\log p_\theta(z_4|z_1,z_2,z_3) - \log q_\phi(z_4|x)]$

$+ \sum_{i=1}^{3} \mathbb{E}_{p(x,t,y)}\mathbb{E}_{q_\phi(z_4|x)}[-KL(q_\phi(z_i|pa_\phi(z_i))||q_\phi(z_i)) - \sum_{j} KL(q_\phi(z_{ij})||p_\theta(z_{ij}|pa(z_{ij})))$

$- KL(q_\phi(z_i)||\prod_{j} q_\phi(z_{ij}))]$

Proof:

$\mathbb{E}_{p(x)p(t|x)p(y|t,x)}[\log p_\theta(x|t,y)]$

$\geq \mathbb{E}_{p(x,t,y)q_\phi(z_1,z_2,z_3,z_4,x|t,y)}[\log p_\theta(x,z_1,z_2,z_3,z_4|t,y) - \log q_\phi(z_1,z_2,z_3,z_4,x|t,y)]$

$\geq \mathbb{E}_{p(x,t,y)q_\phi(z_1,z_2,z_3,z_4,x|t,y)}[\log p_\theta(x|z_4) + \log p_\theta(z_4|z_1,z_2,z_3) + \log p_\theta(z_1|y)$

$+ \log p_\theta(z_2|t,y) + \log p_\theta(z_3|t)$

$- \log q_\phi(z_4|x) - \log q_\phi(z_4|x) - \log q_\phi(z_1|z_4,t) - \log q_\phi(z_2|z_4) - \log q_\phi(z_3|z_4,y)]$

$\geq \mathbb{E}_{p(x,t,y)q_\phi(z_4|x)}[\log p_\theta(x|z_4)] + \mathbb{E}_{q_\phi(z_1,z_2,z_3,x,t,y)}[\log p_\theta(z_4|z_1,z_2,z_3) - \log q_\phi(z_4|x)]$

$+ \mathbb{E}_{p(x,t,y)q_\phi(z_4|x)q_\phi(z_1|z_4,t)}[\log p_\theta(z_1|y) - \log q_\phi(z_1|z_4,t)]$

$+ \mathbb{E}_{p(x,t,y)q_\phi(z_4|x)q_\phi(z_2|z_4)}[\log p_\theta(z_2|t,y) - \log q_\phi(z_2|z_4)]$

$+ \mathbb{E}_{p(x,t,y)q_\phi(z_4|x)q_\phi(z_3|z_4,y)}[\log p_\theta(z_3|t) - \log q_\phi(z_3|z_4,y)]$

$\geq \mathbb{E}_{p(x,t,y)q_\phi(z_4|x)}[\log p_\theta(x|z_4)]$

$- \mathbb{E}_{q_\phi(z_1,z_2,z_3,x,t,y)}[\log q_\phi(z_4|x) - \log q_\phi(z_4) + \log q_\phi(z_4) - \log q_\phi(\prod_{j} z_{4j})$

$+ \log q_\phi(\prod_{j} z_{4j}) - \log p_\theta(z_4|z_1,z_2,z_3)]$

$- \mathbb{E}_{p(x,t,y)q_\phi(z_4|x)q_\phi(z_1|z_4,t)}[\log q_\phi(z_1|z_4,t) - \log q_\phi(z_1) + \log q_\phi(z_1) - \log q_\phi(\prod_{j} z_{1j})$

$+ \log q_\phi(\prod_{j} z_{1j}) - \log p_\theta(z_1|y)]$

$- \mathbb{E}_{p(x,t,y)q_\phi(z_4|x)q_\phi(z_3|z_4,y)}[\log q_\phi(z_2|z_4) - \log q_\phi(z_2) + \log q_\phi(z_2) - \log q_\phi(\prod_{j} z_{2j})$

$+ \log q_\phi(\prod_{j} z_{2j}) - \log p_\theta(z_2|t,y)]$

$- \mathbb{E}_{p(x,t,y)q_\phi(z_4|x)q_\phi(z_3|z_4,y)}[\log q_\phi(z_3|z_4,y) - \log q_\phi(z_3) + \log q_\phi(z_3) - \log q_\phi(\prod_{j} z_{3j})$

$+ \log q_\phi(\prod_{j} z_{3j}) - \log p_\theta(z_3|t)]$

$\geq \mathbb{E}_{p(x,t,y)q_\phi(z_4|x)}[\log p_\theta(x|z_4)]$

$- \mathbb{E}_{q_\phi(z_1,z_2,z_3,x,t,y)}[\log q_\phi(z_4|x) - \log q_\phi(z_4) + \log q_\phi(z_4) - \log q_\phi(\prod_{j} z_{4j})$

$+ \log q_\phi(\prod_{j} z_{4j}) - \log p_\theta(z_4|z_1,z_2,z_3)]$

$- \mathbb{E}_{p(x,t,y)q_\phi(z_4|x)q_\phi(z_1|z_4,t)}[\log \frac{q_\phi(z_1|z_4,t)}{q_\phi(z_1)} + \log \frac{q_\phi(z_1)}{q_\phi(\prod_j z_{1j})} + \sum_j \log \frac{q_\phi(z_{1j})}{p_\theta(z_{1j}|y)}]$

$- \mathbb{E}_{p(x,t,y)q_\phi(z_4|x)q_\phi(z_1|z_4,t)}[\log \frac{q_\phi(z_2|z_4)}{q_\phi(z_2)} + \log \frac{q_\phi(z_2)}{q_\phi(\prod_j z_{2j})} + \sum_j \log \frac{q_\phi(z_{2j})}{p_\theta(z_{2j}|t,y)}]$

$- \mathbb{E}_{p(x,t,y)q_\phi(z_4|x)q_\phi(z_1|z_4,t)}[\log \frac{q_\phi(z_3|z_4,y)}{q_\phi(z_3)} + \log \frac{q_\phi(z_3)}{q_\phi(\prod_j z_{3j})} + \sum_j \log \frac{q_\phi(z_{3j})}{p_\theta(z_{3j}|t)}]$

Writing some of them in explicit KL form, we obtain the ELBO.

**Proposition 1** (Universality of product effect formalization for prognostic score) Let $\mathcal{H}_{\mathcal{X} \times \mathcal{T}}$ be the given Reproducing Kernel Hilbert Space. For any $\epsilon > 0$ and any $f \in \mathcal{H}^n$, there is a $d \in \mathbb{N}$ such that there exist $2n$ d-dimensional function $g_i : \mathcal{X} \to \mathbb{R}^d$ and $h_i : \mathcal{T} \to \mathbb{R}^d$ such that $\|f - (g_i^T h_i, \dots g_n^T h_n)\|_{L_2(P_{\mathcal{X} \times \mathcal{T}})} \leq \epsilon$.

Let $f$ be given, where $f_i \in \mathcal{H}_{\mathcal{X} \times \mathcal{T}}$, and $\mathcal{H}_{\mathcal{X} \times \mathcal{T}}$ is isometrically isomorphic to $\mathcal{H}_{\mathcal{X}} \times \mathcal{H}_{\mathcal{T}}$. Let $\mathcal{H}_{0\mathcal{X} \times \mathcal{T}}$ be the pre-RKHS where $H_{0\mathcal{X} \times \mathcal{T}} = \{f(x,t)|f(x,t) = \sum_{i=1}^{n} \alpha_i k((x_i, t_i), (x, t))\}$ for $1 \leq i \leq n$.

Let $\{f_{in}\} \subseteq H_{0\mathcal{X} \times \mathcal{T}}$ be the associated Cauchy sequence converging to $f_i$. We have that $\{f_{in}\}$ also converges to $f_i$ in $\|\cdot\|_{\mathcal{H}_{\mathcal{X} \times \mathcal{T}}}$ as well.

$\forall \frac{\epsilon^2}{n+1} > 0, \exists N_i > 0$ s.t. $\forall k \geq N_i$, we have $\|f_{ki} - f_i\|_{\mathcal{H}_{\mathcal{X} \times \mathcal{T}}} < \epsilon$.

Let $k' = max(N_1, \dots, N_n)$, then we have $\forall \epsilon > 0$ and a given $f$, $\|f - (f_{k'1}, \dots, f_{k'n})\|_{L_2(P_{\mathcal{X} \times \mathcal{T}})} = (\sum_{i=1}^{n} \int_{\mathcal{X} \times \mathcal{T}} \|f_i - f_{k'i}\|)^{\frac{1}{2}} \leq (n * \frac{\epsilon^2}{n+1})^{\frac{1}{2}} < (n * \frac{\epsilon^2}{n})^{\frac{1}{2}} = \epsilon.$

**Proposition 2** Assume the following hold:

- $f$ and $g$ are injective in Eq.10.

- Let $\psi_\epsilon$ be the characteristic function of $p_\epsilon$. $\{x \in \mathbb{X}|\psi_\epsilon(x) = 0\}$ has measure zero.

- Suppose $z_1 \in \mathbb{R}^a$, $z_2 \in \mathbb{R}^b$, and $z_3 \in \mathbb{R}^c$, $a + b + c = n$, then $\lambda(t, y) = \lambda_1(y) \oplus \lambda_2(t, y) \oplus \lambda_3(t)$, where $\lambda_1(y) \in \mathbb{R}^{2a}$, $\lambda_2(t, y) \in \mathbb{R}^{2b}$, $\lambda_3(t) \in \mathbb{R}^{2c}$ are parameters of gaussian distribution.

- There exists $2n + 1$ points $(t_0, y_0) \dots (t_{2n+1}, y_{2n+1})$ such that the matrix $L = [(\lambda_1(y_1) - \lambda_1(y_0)) \oplus (\lambda_2(t_1, y_1) - \lambda_2(t_0, y_0)) \oplus (\lambda_3(t_1) - \lambda_3(t_0)), \dots, (\lambda_1(y_{2n+1}) - \lambda_1(y_0)) \oplus (\lambda_2(t_{2n+1}, y_{2n+1}) - \lambda_2(t_0, y_0)) \oplus (\lambda_3(t_{2n+1}) - \lambda_3(t_0)))] = [(\lambda(t_1, y_1) - \lambda(t_0, y_0)), \dots, (\lambda(t_{2n+1}, y_{2n+1}) - \lambda(t_0, y_0))]$ is invertible, i.e., $\lambda = \lambda_1 \oplus \lambda_2 \oplus \lambda_3$ where $\lambda_1$ is independent of $t$, and $\lambda_3$ is independent of $y$.

- The sufficient statistics are differentiable almost everywhere.

- Let $k = f \circ g$, then $k(z_1, z_2, z_3) = k_1(z_1) \oplus k_2(z_2) \oplus k_3(z_3) \oplus k_4(z_1, z_2) \oplus k_5(z_1, z_3) \oplus k_6(z_2, z_3) \oplus k_7(z_1, z_2, z_3)$ satisfies $Range(k_i) \cap Range(k_j) = \emptyset$.

then if $p_\theta(x|t, y) = p'_\theta(x|t, y)$ we have

$$k^{-1}(x) = diag(a)k'^{-1}(x) + b. \tag{17}$$

Proof:

$p(x|t, y) \geq p(x'|t', y')$

$\Rightarrow \int p(x, z|t, y)dz = \int p(x', z'|t', y')dz'$

$\Rightarrow \int_{z_1} \int_{z_2} \int_{z_3} \int_{z_4} p(x|z_4)p(z_4|z_1, z_2, z_3)p(z_1|y)p(z_2|t, y)p(z_3|t)$

$= \int_{z_1'} \int_{z_2'} \int_{z_3'} \int_{z_4'} p(x'|z_4')p(z_4'|z_1', z_2', z_3')p(z_1'|y')p(z_2'|t', y')p(z_3'|t')dz'$

$\Rightarrow \int_{\mathcal{X}} p_\epsilon(x - \bar{x})p(g^{-1} \circ f^{-1}(\bar{x})|t, y) vol\ J_{f^{-1}}(\bar{x}) vol\ J_{g^{-1}}(f^{-1}(\bar{x}))d\bar{x}$

$= \int_{\mathcal{X}} p_\epsilon(x - \bar{x}')p(g^{-1} \circ f^{-1}(\bar{x}')|t, y) vol\ J_{f^{-1}}(\bar{x}') vol\ J_{g^{-1}}(f^{-1}(\bar{x}'))d\bar{x}'$

$\Rightarrow (\tilde{p}_{z_1, z_2, z_3, z_4} * p_\epsilon)(x) = (\tilde{p}_{\bar{z}_1, \bar{z}_2, \bar{z}_3, \bar{z}_4} * p_\epsilon)(x)$

$\Rightarrow F[\tilde{p}_{z_1, z_2, z_3, z_4}](w)\psi_\epsilon(w) = F[\tilde{p}_{\bar{z}_1, \bar{z}_2, \bar{z}_3, \bar{z}_4}](w)\psi_\epsilon(w)$

$\Rightarrow \tilde{p}_{z_1, z_2, z_3, z_4}(x) = \tilde{p}_{\bar{z}_1, \bar{z}_2, \bar{z}_3, \bar{z}_4}(x)$

$$\Rightarrow log\ vol\ J_{f^{-1}}(x) + log\ vol\ J_{g^{-1}}(f^{-1}(x))$$

$$+ \sum_{i=1}^{a}(log\ Q_i((g^{-1}\circ f^{-1})_i(x)) - log\ Z_3(t) + \sum_{j=1}^{2}T_{i,j}((g^{-1}\circ f^{-1})_i(x))\lambda_{i,j}(t))$$

$$+ \sum_{i=a+1}^{a+b}(log\ Q_i((g^{-1}\circ f^{-1})_i(x)) - log\ Z_3(t,y) + \sum_{j=1}^{2}T_{i,j}((g^{-1}\circ f^{-1})_i(x))\lambda_{i,j}(t,y))$$

$$+ \sum_{i=a+b+1}^{n}(log\ Q_i((g^{-1}\circ f^{-1})_i(x)) - log\ Z_3(y) + \sum_{j=1}^{2}T_{i,j}((g^{-1}\circ f^{-1})_i(x))\lambda_{i,j}(y))$$

$$= log\ vol\ J_{\tilde{f}^{-1}}(x) + log\ vol\ J_{\tilde{g}^{-1}}(\tilde{f}^{-1}(x))$$

$$+ \sum_{i=1}^{a}(log\ \tilde{Q}_i((\tilde{g}^{-1}\circ \tilde{f}^{-1})_i(x)) - log\ \tilde{Z}_3(t) + \sum_{j=1}^{2}\tilde{T}_{i,j}((\tilde{g}^{-1}\circ \tilde{f}^{-1})_i(x))\tilde{\lambda}_{i,j}(t))$$

$$+ \sum_{i=a+1}^{a+b}(log\ \tilde{Q}_i((\tilde{g}^{-1}\circ \tilde{f}^{-1})_i(x)) - log\ \tilde{Z}_i(t,y) + \sum_{j=1}^{2}\tilde{T}_{i,j}((\tilde{g}^{-1}\circ \tilde{f}^{-1})_i(x))\tilde{\lambda}_{i,j}(t,y))$$

$$+ \sum_{i=a+b+1}^{n}(log\ \tilde{Q}_i((\tilde{g}^{-1}\circ \tilde{f}^{-1})_i(x)) - log\ \tilde{Z}_i(y) + \sum_{j=1}^{2}\tilde{T}_{i,j}((\tilde{g}^{-1}\circ \tilde{f}^{-1})_i(x))\tilde{\lambda}_{i,j}(y)).$$

Let the $2n+1$ points in the assumption be given. Define $\bar{\lambda}(t,y) = \lambda_1(y) - \lambda_1(y_0) \oplus \lambda_2(t,y) - \lambda_2(t_0,y_0) \oplus \lambda_3(t) - \lambda_3(t_0)$. Subtracting the equations for $(t_i, y_i)$ from the equation for $(t_0, y_0)$, we get

$$< T(g^{-1}\circ f^{-1}(x)), \bar{\lambda}(t_k, y_k) > + \sum_i \log \frac{Z_3(t_0,y_0)}{Z_3(t_k,y_k)} + \sum_i \log \frac{Z_3(t_0)}{Z_3(t_k)} + \sum_i \log \frac{Z_3(y_0)}{Z_3(y_k)}$$

$$= < \tilde{T}(\tilde{g}^{-1}\circ \tilde{f}^{-1}(x)), \bar{\tilde{\lambda}}(t_k, y_k) > + \sum_i \log \frac{\tilde{Z}_i(t_0,y_0)}{\tilde{Z}_i(t_k,y_k)} + \sum_i \log \frac{\tilde{Z}_i(t_0)}{\tilde{Z}_i(t_k)} + \sum_i \log \frac{\tilde{Z}_i(y_0)}{\tilde{Z}_i(y_k)}.$$

Define $b_k = \sum_i \log \frac{Z_3(\tilde{t_0},y_0)Z_3(t_k,y_k)}{Z_3(t_0,y_0)\tilde{Z}_3(t_k,y_k)} + \sum_i \log \frac{Z_3\tilde{(t_0)}Z_3(t_k)}{Z_3(t_0)\tilde{Z}_3(t_k)} + \sum_i \log \frac{Z_3\tilde{(y_0)}Z_3(y_k)}{Z_3(y_0)\tilde{Z}_3(y_k)}.$

Arranging the terms, we get

$$L^T T(g^{-1}\circ f^{-1}(x)) = \tilde{L}^T T(\tilde{g}^{-1}\circ \tilde{f}^{-1}(x)) + b.$$

Hence we have

$$T(g^{-1}\circ f^{-1}(x)) = AT(\tilde{g}^{-1}\circ \tilde{f}^{-1}(x)) + c.$$

Denote $h = f \circ g$, we then have $T(h^{-1}(x)) = AT(\tilde{h}^{-1}(x)) + c$.

Hence, by following the same line of reasoning as Sorrenson et al. (2020), we have that $k^{-1}(x) = diag(a)k'^{-1}(x) + b$

**Theorem 1** Suppose **Assumption 4.1** - **Assumption 4.4** hold. Furthermore, $\tilde{K}_i$ and $K_i$ are injective for all $i$. Then if $\mathbb{E}_{p_\theta}[X|Z_1, Z_2, Z_3] = \mathbb{E}[X|\tilde{Z}_1, \tilde{Z}_2, \tilde{Z}_3]$, we have:

1. (Recovery of latent code) If either
   1) $\tilde{K}_1$, $\tilde{K}_2$ and $\tilde{K}_3$ are not empty mapping, or
   2) at least two of $\tilde{K}_4$-$\tilde{K}_7$ are non-empty mappings, $I(\Delta_T \tilde{Z}_1; T) = 0$, $I(\Delta_Y \tilde{Z}_3; Y|T) = 0$ for some injective $\Delta_T$ and $\Delta_Y$, $I(Z_2; T) \neq 0$ and $I(Z_2; Y) \neq 0$,
   then $Z_1 = \Delta_1 \circ \tilde{Z}_1$, $Z_2 = \Delta_2 \circ \tilde{Z}_2$, $Z_3 = \Delta_3 \circ \tilde{Z}_3$ for some injective mapping $\Delta_1$, $\Delta_2$, $\Delta_3$.

2. (Recovery of bPGS via subset of covariates) $Z = Z_1 \oplus Z_2 = v \circ p$ for some injective mappint $v$. Moreover, the overlapping condition can be relaxed onto

$X' \subseteq X$ where where $\mathcal{X}' := \{x \in \mathcal{X} | k_4^{*-1}(x) \text{ is overlapping}\} \cup \{x \in \mathcal{X} | k_1^{*-1}(x) \text{ and } k_2^{*-1}(x) \text{ is overlapping}\} \cup \{x \in \mathcal{X} | k_7^{*-1}(x) \text{ is overlapping}\}$.

3. (OOD generalization on non-overlapping regions) Suppose $\tilde{f}_t(x) = \mathbb{E}[Y|X,T] = E_{p_\theta}[Y|X,T] = f_t(x)$ for all observed $(x,t) \in \mathcal{X} \times \mathcal{T}$. Suppose $\exists t' \in \mathcal{T}$ s.t. $j'_t$ and $\tilde{j}'_t$ are injective. Suppose there exist a RKHS $\mathcal{H}_\mathcal{P}$ on the bPGS space, also $\tilde{j}^*_t \in \mathcal{H}_\mathcal{P}$ and $j^*_{t^*} \circ \Delta \in \mathcal{H}_\mathcal{P}$ for all $t^* \in \mathcal{T}$ where $\Delta := j'^{-1}_t \circ \tilde{j}'_t$. Then we have $||j_t \circ \Delta - \tilde{j}_t|| < \epsilon \Rightarrow |\tilde{f}_t(x) - f_t(x)| < \epsilon * C$ for some constant $C$ for all $t \in \mathcal{T}$.

Since we have $\mathbb{E}_{p_\theta}[X|Z_1, Z_2, Z_3] = \mathbb{E}[X|\tilde{Z}_1, \tilde{Z}_2, \tilde{Z}_3]$ and that $\tilde{K}_1, \tilde{K}_2, \tilde{K}_3$ are non-empty mapping, we have $\tilde{K}_1(\tilde{Z}_1) = K_1(Z_1), \tilde{K}_2(\tilde{Z}_2) = K_2(Z_2), \tilde{K}_3(\tilde{Z}_3) = K_3(Z_3)$. Since $\tilde{K}_i$ and $K_i$ are injective for all $i$, $Z_1 = K_1^{-1}\tilde{K}_1(\tilde{Z}_1), Z_2 = K_2^{-1}\tilde{K}_2(\tilde{Z}_2), Z_3 = K_3^{-1}\tilde{K}_3(\tilde{Z}_3)$, hence the result.

We will show the case where $\tilde{K}_4$ and $\tilde{K}_6$ are non-empty, other cases are similar. Since we have $K_4(Z_1 \oplus Z_2) = \tilde{K}_4(\tilde{Z}_1 \oplus \tilde{Z}_2), Z_1 \oplus Z_2 = K_4^{-1}\tilde{K}_4(\tilde{Z}_1 \oplus \tilde{Z}_2)$. Consider $K_4^{-1}\tilde{K}_4(\tilde{Z}_1)$. Let $\Delta_T := K_4^{-1}\tilde{K}_4$. Since we have $I(K_4^{-1}\tilde{K}_4(\tilde{Z}_1); T) = 0$ and $I(Z_2; T) \neq 0$, it then follows that $K_4^{-1}\tilde{K}_4(\tilde{Z}_1) \cap Z_2 = \emptyset$, hence $K_4^{-1}\tilde{K}_4(\tilde{Z}_1) \subseteq Z_1$. The other side of set inclusion can be similarly shown, hence $K_4^{-1}\tilde{K}_4(\tilde{Z}_1) = Z_1$. Similarly, we have $Z_2 \oplus Z_3 = K_6^{-1}\tilde{K}_6(\tilde{Z}_2 \oplus \tilde{Z}_3)$. Consider $K_6^{-1}\tilde{K}_6(\tilde{Z}_3)$. Let $\Delta_Y := K_6^{-1}\tilde{K}_6$. Since we have $I(K_6^{-1}\tilde{K}_6\tilde{Z}_3; Y|T) = 0$ and $I(Z_2; Y|T) \neq 0$, it then follows that $K_6^{-1}\tilde{K}_6(\tilde{Z}_3) \cap Z_2 = \emptyset$, hence $K_6^{-1}\tilde{K}_6(\tilde{Z}_3) \subseteq Z_3$, and the other side of set inclusion can be similarly shown. By set exclusion we also have $K_4^{-1}\tilde{K}_4(\tilde{Z}_2) = Z_2$.

We will show the case for $\{x \in \mathcal{X} | k_4^{*-1}(x) \text{ is overlapping}\}$. The other cases shall be similar. Let $\Delta : \mathcal{P} \to \mathbb{R}^n$ be an injective mapping. Let $z(x) = k_4^{*-1}(x) = z^* = \Delta \circ p$, then clearly $k_4^{*-1}$ is a set of optimal parameters. According to **Proposition 2**, we have $z = k_4^{-1} = A \circ k_4^{*-1} = A \circ \Delta \circ p$. Let $v := A \circ \Delta$, hence the result.

Let such $t'$ be given. $\tilde{j}'_t \circ p = j'_t \circ z \Rightarrow z = j'^{-1}_t \circ \tilde{j}'_t \circ p$ since $j_t$ is injective. $\forall x$, let its corresponding $p(x)$, denoted hereafter as $p$, be given. $\forall t \in T$,

we then have $||j_t \circ z - \tilde{j}_t \circ p|| = ||j_t \circ j'^{-1}_t \circ \tilde{j}'_t - \tilde{j}_t \circ p|| \leq ||\delta_p|| ||j_t \circ \Delta - \tilde{j}_t|| < \epsilon * C$.

Note that the overlapping condition is significantly relaxed. By modeling the data generating process for $\mathcal{X}$, we only require the overlapping condition to hold on covariates resulting from their generating latent factors, which can potentially be a lower dimensional space depending on the dataset. Also, once a bPGS is learned up to an invertible transformation, we can always utilize the information on any treatment arm $j_t$ to extrapolate beyond the seen regions.

## B    EXPERIMENTAL DETAILS

### B.1    BINARY SYNTHETIC DATASET

**Dataset.**    We generate the synthetic dataset in the following way:

- Generate $n$ samples according to $N(\mu_v, \sigma_v)$ for $v \in \{Z_1, Z_2, Z_3\}$.
- Sample the prognostic coefficient $P$ from $N(\mu_{prog}, \sigma_{prog})$.
- Sample the balancing coefficient $B$ from $N(\mu_{bal}, \sigma_{bal})$.
- Let $Z_p = Z_1 \oplus Z_2$ be the prognostic score, and $Z_b = Z_2 \oplus Z_3$ be the balancing score.
- Generate the outcome as $Y_t \sim N(f_T(Z_p)P, 0.1)$.
- Generate the treatment as $T \sim Bern(\omega(Z_b B))$, where $\omega$ controls the non-overlapping level.

**Hyperparameter Search Range**    To facilitate hyperparameter search, we integrate the ASHAScheduler (Li et al., 2020) in ray tune (Moritz et al., 2018) into our training work flow. The corresponding api call used for each hyperparameter is listed in Table 3, Table 4, and Table 5.

Table 3: Intact VAE Hyperparameter Search Range.

| Hyperparameters | Search Range |
|---|---|
| Beta Coefficient | loguniform(1e-2, 1e2) |
| Number of Encoder Layers | choice([1, 2, 3, 4]) |
| Encoder Layer Size | qrandint(20, 100, 5) |
| Decoder Layer Size | qrandint(20, 100, 5) |
| Latent Layer Size | qrandint(5, 50, 5) |
| Number of Outcome Layers | choice([1, 2, 3, 4] |
| Outcome Layer Size | qrandint(20, 100, 5) |

Table 4: DR-CFR Hyperparameter Search Range.

| Hyperparameters | Search Range |
|---|---|
| adjustment layer size | qrandint(10, 100, 10) |
| Number of adjustment layers | qrandint(2, 3, 1) |
| adjustment hidden layer size | qrandint(10, 100, 10) |
| confounder layer size | qrandint(10, 100, 10) |
| Number of confounder layers | qrandint(2, 3, 1) |
| confounder hidden layer size | qrandint(10, 100, 10) |
| instrument layer size | qrandint(10, 100, 10) |
| Number of instrument layers | qrandint(2, 3, 1) |
| instrument hidden layer size | qrandint(10, 100, 10) |
| Number of outcome layers | qrandint(2, 4, 1) |
| Outcome hidden layer size | qrandint(50, 200, 10) |
| Imbalance coefficient | loguniform(1e-1, 1e1) |
| treatment loss coefficient | loguniform(1e-1, 1e1) |

## B.2 STRUCTURED DATASET

**Covariates.** For covariates, we use the gene expression measurements from *The Cancer Genomic Atlas Simulation* (Weinstein et al., 2013), which is used in Kaddour et al. (2021)'s work as well. We use Kaddour et al. (2021)'s preprocessing implementation, with 4000/1000/4659 units in the train/validation/test datasets, respectively.

**Treatment Graphs.** For treatment graphs, we use the tox21 dataset from the MoleculeNet (Wu et al., 2018) as our structured treatments. To conduct scaffold split over the treatments, we adopt the preprocessing implementation used by Hu et al. (2019). This setup presents a greater challenge compared to the approach used in Kaddour et al. (2021), where random splitting over treatment graphs is employed. As such, we are testing the model's ability to generalize beyond *seen* and *in-distribution* treatments. In his influential work, Pearl (2009) states that *"Causal models should therefore be chosen by a criterion that challenges their stability against changing conditions..."* (also discussed in the introduction of Shalit et al. (2017)). Hence, our curated setting can be viewed as an extreme means to rigorously evaluate the model's stability.

**Data Generation Process.** We use a similar outcome generation scheme as that of Kaddour et al. (2021). Given a 12-dimensional molecular properties $z \in \mathbb{R}^{12}$, and covariate $\mathcal{X}$, the outcome is generated as:

$$Y = \frac{1}{3}\mu_0(x) + \frac{9}{10}z^T x^{(PCA)} + \epsilon, \quad \epsilon \sim N(0, 1).$$

**Hyperparameter Search Range** To facilitate hyperparameter search, we integrate the ASHAScheduler (Li et al., 2020) in ray tune (Moritz et al., 2018) into our training work flow. The corresponding api call used for each hyperparameter is listed in Table 6 and Table 7.

Table 5: DIRE Hyperparameter Search Range.

| Hyperparameters | Search Range |
|---|---|
| $Z_1$ layer size | qrandint(10, 100, 10) |
| Number of $Z_1$ layers | qrandint(2, 3, 1) |
| $Z_2$ layer size | qrandint(10, 100, 10) |
| Number of $Z_2$ layers | qrandint(2, 3, 1) |
| $Z_3$ layer size | qrandint(10, 100, 10) |
| Number of $Z_3$ layers | qrandint(2, 3, 1) |
| $Z_4$ layer size | qrandint(10, 100, 10) |
| Number of $Z_4$ layers | qrandint(2, 3, 1) |
| Number of outcome layers | qrandint(2, 4, 1) |
| Outcome hidden layer size | qrandint(50, 200, 10) |
| Imbalance coefficient | loguniform(1e-1, 1e1) |
| treatment loss coefficient | loguniform(1e-1, 1e1) |
| $Z_1$ mutual info | qrandint(0.1, 20) |
| $Z_2$ mutual info | qrandint(0.1, 20) |
| $Z_3$ mutual info | qrandint(0.1, 20) |
| $Z_4$ mutual info | qrandint(0.1, 20) |
| $Z_1$ total correlation | loguniform(1e-1, 1e1) |
| $Z_2$ total correlation | loguniform(1e-1, 1e1) |
| $Z_3$ total correlation | loguniform(1e-1, 1e1) |
| $Z_4$ total correlation | loguniform(1e-1, 1e1) |
| prior coefficient | uniform(0.1, 1.0) |
| $Z_1$ decoder layer size | qrandint(5, 20, 5) |
| $Z_2$ decoder layer size | qrandint(5, 20, 5) |
| $Z_3$ decoder layer size | qrandint(5, 20, 5) |
| Number of $Z_1$ decoder layers | qrandint(1, 2, 1) |
| Number of $Z_2$ decoder layers | qrandint(1, 2, 1) |
| Number of $Z_3$ decoder layers | qrandint(1, 2, 1) |
| Number of reconstruction hidden layers | qrandint(2, 3, 1) |
| Reconstruction layer size | qrandint(20, 100, 10) |
| ELBO coefficient | loguniform(1e-1, 1e1) |

**Model and Implementation.** To better evaluate if DIRE learns a better balanced prognostic score, we incorporate DIRE with Generalized Robinson Decomposition (Kaddour et al., 2021). The Generalized Robinson Decomposition for the outcome is:

$$Y = m_\phi(Z_1, Z_2) + g_\eta(Z_1, Z_2)^T(h(T) - e_\psi(Z_2)) + \epsilon.$$

Importantly, the value of $m_\phi$ is solely determined by adjustments and confounders, while $g_\eta$ and $e_\psi$ depend only on confounders. We keep the $h(T)$ representation fixed (we also provide the same information to SIN-with-Aux-Info) and utilize the corresponding latent representations in the downstream plug-in estimators. We also adopt a two-stage learning process as in Kaddour et al. (2021):

**Stage 1.** We learn the mean outcome model using latent factors relevant to $\mathcal{Y}$. Denote the regularizers as $\Lambda(\cdot)$, the loss of the mean outcome model is:

$$\mathcal{L}_{\phi,\theta} = \frac{1}{m}(y_i - m_{\phi,\theta}(Z_1, Z_2))^2 + \mathcal{L}_{ELBO} + \Lambda(\phi) + \Lambda(\theta). \tag{18}$$

**Stage 2.** In contrast to the approach taken by Kaddour et al. (2021), we learn $m_\phi, g_\eta, e_\psi$ in this stage, fixing the treatment representation. The loss in this stage is derived as:

$$\mathcal{L}_{\eta,\psi,\theta,\phi} = \frac{1}{n}(y_i - g_\eta(Z_2))^2 + \alpha_\psi \frac{1}{n}(t_i - e_\psi(Z_2))^2 + \alpha_\phi \frac{1}{n}(y_i - m_{\phi,\theta}(Z_1, Z_2))^2,$$
$$+ \mathcal{L}_{ELBO} + \Lambda(\eta) + \Lambda(\psi) + \Lambda(\theta) + \Lambda(\phi). \tag{19}$$

where $\alpha_\psi, \alpha_\phi$ are hyperparameters.

Table 6: SIN structured Hyperparameter Search Range.

| Hyperparameters | Search Range |
|---|---|
| Output dimension of $g(X), h(T)$ | qrandint(2, 800) |
| Dimension of hidden covariates layer | qrandint(100, 400) |
| Number of Update Steps of Global Objective | qrandint(10, 20) |
| Dimension of hidden conditional mean outcome layers | qrandint(100, 300) |
| Dimension of hidden propensity layers | qrandint(10, 80, 1) |
| Number of covariates layers | qrandint(2, 4) |
| Number of conditional mean outcome layers | qrandint(2, 4) |
| Number of propensity layers | qrandint(1, 4, 1) |
| patience | {30} |
| batch size | {500} |
| Propensity Net Weight Decay | {0.0, 1e-4} |
| Propensity Net Learning Rate | { 1e-3, 5e-4 } |
| Conditional Mean Outcome Net Learning Rate | { 1e-3, 5e-4 } |
| Conditional Mean Outcome Net Weight Decay | { 0.0, 1e-4 } |
| GNN Learning Rate | { 1e-3, 5e-4 } |
| GNN Weight Decay | { 0.0, 1e-4 } |
| Covariates Net Learning Rate | { 1e-3, 5e-4 } |
| Covariates Net Weight Decay | { 0.0, 1e-4 } |
| Max Epochs | { 400 } |

Table 7: DIRE structured Hyperparameter Search Range.

| Hyperparameters | Search Range |
|---|---|
| Outcome Discretization dimension | qrandint(5, 20, 1) |
| ELBO coefficient | loguniform(1e-2, 1e2) |
| $Z_1$ mutual info | uniform(0.1, 25.0) |
| $Z_2$ mutual info | uniform(0.1, 25.0) |
| $Z_3$ mutual info | uniform(0.1, 25.0) |
| $Z_4$ mutual info | uniform(0.1, 25.0) |
| $Z_1$ total correlation | uniform(0.1, 25.0) |
| $Z_2$ total correlation | uniform(0.1, 25.0) |
| $Z_3$ total correlation | uniform(0.1, 25.0) |
| $Z_4$ total correlation | uniform(0.1, 25.0) |
| $Z_1$ prior coeff | uniform(0.1, 1.0) |
| $Z_2$ prior coeff | uniform(0.1, 1.0) |
| $Z_3$ prior coeff | uniform(0.1, 1.0) |
| $Z_4$ prior coeff | uniform(0.1, 1.0) |
| $Z_1$ decoder layer size | qrandint(5,50, 1) |
| $Z_2$ decoder layer size | qrandint(5,50, 1) |
| $Z_3$ decoder layer size | qrandint(5,50, 1) |
| Number of $Z_1$ decoder layers | qrandint(1, 4, 1) |
| Number of $Z_2$ decoder layers | qrandint(1, 4, 1) |
| Number of $Z_3$ decoder layers | qrandint(1, 4, 1) |
| Reconstruction Layer Size | qrandint(20, 200, 1) |
| Number of reconstruction layers | qrandint(2, 4, 1) |
| Conditional Mean Outcome Model coefficient | uniform(0.1, 1.0) |
| VAE Learning Rate | { 1e-3, 5e-4 } |
| VAE Weight Decay | { 0.0, 1e-4 } |

---

**Algorithm 1** SIN with DIRE

---

**Input:** Data $\{(x_i, t_i, y_i)\}$, stage 1 batch size $m$, stage 2 batch size $n$, step size $\lambda_\eta, \lambda_\phi, \lambda_\psi, \lambda_\theta, \lambda_\phi$
   Initialize $\eta, \phi, \psi, \theta, \phi$
   **while** not coverged **do**
      Sample mini-batch $\{x_i, y_i\}_{i=1}^m$
      Evaluate $\mathcal{L}_{\phi,\theta}$ in Eq.18
      $\phi \leftarrow \phi - \lambda_\phi \hat{\nabla}_\phi \mathcal{L}_{\phi,\theta}$
      $\theta \leftarrow \theta - \lambda_\theta \hat{\nabla}_\theta \mathcal{L}_{\phi,\theta}$
   **end while**
   **while** not converged **do**
      Sample mini-batch $\{x_i, y_i\}_{i=1}^n$
      Evaluate $\mathcal{L}_{\eta,\psi,\theta,\phi}$ in Eq.19
      $\phi \leftarrow \phi - \lambda_\phi \hat{\nabla}_\phi \mathcal{L}_{\phi,\theta}$
      $\psi \leftarrow \psi - \lambda_\psi \hat{\nabla}_\psi \mathcal{L}_{\eta,\psi,\theta,\phi}$
      $\theta \leftarrow \theta - \lambda_\theta \hat{\nabla}_\theta \mathcal{L}_{\eta,\psi,\theta,\phi}$
      $\eta \leftarrow \eta - \lambda_\eta \hat{\nabla}_\eta \mathcal{L}_{\eta,\psi,\theta,\phi}$
   **end while**

---

