# OpenReview forum: "Learning Identifiable Balanced Prognostic Score for Treatment Effect Estimation Under Limited Overlap"
_ICLR.cc/2024/Conference — Submitted to ICLR 2024_

### Official Review · Reviewer_DxND · 2023-10-17

**Soundness:** 1 poor
**Presentation:** 1 poor
**Contribution:** 2 fair
**Rating:** 3
**Confidence:** 4

**Summary:**

**Post-rebuttal update**:
I am sorry, but I have to say most main claims of the paper still look a mess to me. I maintain my score but add my final reply to the author(s).

Markov condition. "This update on assumption will not impact the fundamental methodology of our proof." I am dubious. Could you highlight how this assumption is used in the proof?

Injective is unrealistic. I do not understand what you mean by "their dimensions can be hyperparameterized". Anyway, could you show that, for function g, dim(image) ≥ dim(domain)?

Assum 4.2. My concern was just that the second equality seems trivially satisfied. Then, why it is an assumption?

**End update**


The paper proposes a new identifiable VAE to disentangle and identify the instrumental variable, hidden confounder, and prognostic score. Theoretical analysis and experimental results are provided.

**Strengths:**

Using identifiable VAE, and more generally deep identifiable models, to estimate causal effect is a promising recent direction.

It is interesting to see the potential of identifiable VAE to handle zero overlap.

Experiments show favorable performance regarding estimation accuracy.

**Weaknesses:**

*Impossible/unrealistic theoretical assumptions, and (almost) assume the main results*

~~Most importantly, Theorem 1 result 1 directly assumes the 3 hidden variables are identified up to injective mappings, but this is what we want to prove! For example, this is the major goal of the theoretical analysis in the Intact-VAE paper.~~
Moreover, the 2nd independence assumption (through mutual information) violates the Markov condition; Z3 and Y are related through the path Z3-T-Y. Any violations of Markov condition are dubious without detailed justification.

In Prop 2, g maps three variables to a variable. Then, injectivity is impossible unless the domain of g is in fact a 1-dim manifold embedded in the 3-dim space. Similarly, assuming functions K4-K7 are injective is unrealistic.

*Insufficient experimental evidence, particularly for identification and zero overlap*

The experiments only examine the ATE error and PEHE, without touching on identification. In Fig 3, there are no legends for the x-axis, but the tendency of the proposed method is no better than others.


In Sec 5.3, the claim “even in the in-sample scenario, β-Intact-VAE struggles to generate a balanced prognostic score in the presence of instruments” is unconvincing. You only show your method has lower errors, but the reason might be, for example, that your method has better fitting capacity. Moreover, the claim “The performance of DR-CFR diminishes as the limited overlapping level becomes more severe” is also unconvincing, the differences are very small and in the range of error bars.

*The claim of handling zero overlap is not theoretically supported.*

But the claims in the Abstract and contributions make readers think the opposite. The statement “since DIRE also generalizes its identification capability to the out-sample setting, …” at the end of Sec 5.2 is also read as if there are theoretical supports.

*No support for the identification of balancing score*. In fact, after it is claimed in the Abstract and contributions, the only place this is mentioned is at the end of Sec 4 “we predict the treatment using the identifiable balancing score”; again, a blank claim.

*Discrepancy between the theory and method*

If the theory works, there is no need to use the “ELBO decomposition trick” and to add L_{prognostic score} and L_{balancing score}. In general, any departure of practical choices and the theory should be discussed and/or supported by experiments.

*Writing is very unclear and sloppy*

The theoretical assumptions and statements are not clear and/or not discussed clearly.

- Assum 4.1, the “circle-plus” symbol is used without introduction. I assume it means “concatenate the dimensions together”.
- Assum 4.2, I assume p in eq3 means p(Z1, Z2), but, if j_t is just a general function, what is the difference between the two sides of the 2nd equality? I don’t think “the second equality is obtained through backdoor criterion” is a meaningful explanation. Also, the symbol j_t is used without introduction (though I know you are following the previous work.)
- Assum 4.4 comes from nowhere. We need the discussion of its causal meaning, or, if it is a technical assumption, why it should hold in practice should be discussed. For example, rank conditions are usually critical for identification, so we care what is n here?
- I cannot understand Prop 1. There is a “where j =” clause, but j is even not mentioned before! (I assume it is not j_t in eq4). And I cannot understand why it means “we can always derive a balanced prognostic score”.
- In Theorem 1, p_{theta} is used before introduction.
- Symbol p is overloaded, meaning either prognostic score or probability.

The major (unrealistic) assumptions, as I mentioned in the first weakness, are not listed formally as in Assumption 4.1 etc, but are mentioned in the pass in the theorems. This is very sloppy writing.

Figure 3 is not mentioned in the main text.

**Questions:**

Please address the issues/questions raised in the Weaknesses.

My general suggestion is that, don’t take hidden confounding lightly! Causal effect identification and estimation under hidden confounding is an extremely hard problem. I do not want to over-generalize, but I have not seen a single conference paper that rigorously addresses this problem under non-standard settings. I refer to standard settings as, for example, IVs, proxy variables, balancing scores, prognostic scores (but usually not combinations of them), and, in general, those studied in the causal inference literature and published at Biometrika, Econometrica, Journal of Econometrics, Journal of the American Statistical Association, Journal of the Royal Statistical Society Series B, Annal of Statistics, etc. If you check the Intact-VAE paper, which is your main reference, you will find it indeed refers to those journals a lot.

---

> ### Author Response · Authors · 2023-11-15
> **Response for the concerns (part 1).**
>
> Thanks for your reviews and comments. We believe there exist some misunderstandings regarding our work. Firstly, we would like to clarify that our primary motivation is to tackle the limited overlapping problem by assuming unconfoundedness, rather than focusing on the hidden confounding problem. Furthermore, we want to emphasize that certain results have been rigorously proven in the appendix of our work, rather than being solely assumed. These results have been derived using reasonable and more relaxed assumptions compared to previous studies. We will provide explanations for all the questions mentioned in the following.
>
> **Q1: Hidden variables in Theorem 1.**
>
> We would like to clarify that we have not made the assumption that the three latent variables are identified up to injective mappings. On the contrary, we have provided a proof to support this assertion. For details, please refer to Appendix.
>
> **Q2: The MI assumption violates the Markov condition.**
>
> Thanks for raising your concerns. We want to clarify that the Markov condition is indeed not violated by the MI assumption. The conditional independence of Y and Z3 given T can be satisfied due to the chain structure present in our model. It is important to note that this holds true when using only the local Markov assumption, which has been demonstrated and proven in [3]. To be mentioned, the MI assumption/technique is commonly used in recent literature [4-6], and we will highlight this in the revised version of our work.
>
> **Q3: Assumption that some functions are injective is unrealistic.**
>
> I would like to address a potential misunderstanding from the reviewer. We want to clarify that we have not made any specific assumptions regarding the dimensions of Z1, Z2, Z3, and Z4 in our work. In fact, their dimensions can be hyperparameterized based on the specific requirements of the problem. These variables are learned as embeddings, allowing for flexibility in their dimensions, which is common in our community [4-6]. As such, our injectivity assumption directly inherits from [2], where such assumption is also made in [1].
>
> **Q4: The experiments do not touch on identification. No legends for the x-axis, in Fig 3.**
>
> Thank you for bringing to our attention the oversight regarding the missing legends for the x-axis. We will address this issue and correct any typos in our future revisions. To clarify, the legends for the x-axis are the same as those used in [8]. They represent the number of treatments considered, specifically PEHE@K, which is the metric introduced in paragraph 2 of section 5.4. It is worth noting that the error bars of our proposed models are significantly lower than the rest, indicating their statistical significance, especially when generalizing to more long-tail treatments.
>
> **Q5: Unconvincing claim in Sec 5.3.**
>
> **Claim of β-Intact-VAE.** First, β-Intact-VAE clearly performs worse than the sanity-checking baseline ZERO that consistently predicts zero casual effect when omega=15.0, indicating that it fails to learn meaningful causal effect.
> As we move from left to right in the figure, the degree of limited overlapping increases, and our PEHE shows a decreasing trend. Besides, the error bars of DIRE clearly do not overlap with that of DR-CFR when omega is 10.0 and 15.0. Furthermore, for fair comparison, the experiment setting of all methods (including the number of hyperparameter searches) are the same which can be deemed as the equal fitting capacity to some extent. In addition, based on assumption 4.2, we can understand why a better identification of a balanced prognostic score leads to more accurate treatment effect estimation. Since the outcome is a mapping from the balanced prognostic score to the outcome, and its empirical realization is an MLP with similar (or identical) predictive capacity. Consequently, the accuracy of y reflects the quality of p, which in turn implies the quality of identification.
>
> **Claim of DRCFR.** The differences is not mall, on the contrary, it is huge. Specifically, when omega = 10.0, DR-CFR’s out-sample pehe value is 3.65±0.15, whereas DIRE’s pehe value is 3.33±0.12. When omega = 15.0, DR-CFR’s out-sample pehe value is 3.74±0.15, whereas DIRE’s pehe value is 3.30±0.13. DR-CFR’s pehe error is increasing while the degree of limited overlap increases, whereas DIRE’s performance remains robust when confronted with extreme limited overlap.
>
> **Q6: No support for the identification of balancing score.**
>
> We have the support for the identification, and the reasons are as follows. Since we have provided proof of identification for latent instruments and confounders, and the balancing score is naturally realized as the concatenation of these variables, the identifiability of the balancing score naturally follows.

---

> ### Author Response · Authors · 2023-11-15
> **Response for the concerns (part 2).**
>
> **Q7: No need to use the “ELBO decomposition trick” and to add $L_{prognostic score}$ and $L_{balancing score}$.**
>
> I am afraid that I can not agree with the reviewer. Actually, the ELBO decomposition trick plays a crucial role in learning disentangled factors of variation to fulfill the underlying theory. Specifically, in theorem 2, the mutual information term is responsible for minimizing the mutual information between Z1/T and Z3/Y, which aligns with the assumption stated in theorem 1 result 1. The total correlation term is designed to learn representations that adhere to independent causal mechanisms [10]. When combined with our encoder/decoder structure, the ELBO decomposition trick provides a critical and elegant approach to learning disentangled and identifiable latent representations for the underlying data generation process. It allows us to extract meaningful information and capture the causal relationships within the data. Moreover, $L_{prognostic score}$ and $L_{balancing score}$ are commonly utilized in various disentanglement-based approaches[4-6,9] for treatment effect estimation, which obviously contributes to learn the disentangled representation.
>
> **Q8: Assum 4.4 comes from nowhere.**
>
> Thanks for your concerns again. Assumption 4.4 can be seen as a generalization of assumption 3 in [8]. In [8], The outcome is parameterized as $Y = g(X)^T h(T) + \epsilon$. In our assumption, we consider $Y$ itself to be a prognostic score as it serves as a sufficient statistic for the outcome. Additionally, $g(X)$ is a balanced prognostic score since $Y$ is conditionally independent of $X$ given $g(X)$. To extend this assumption, we allow $Y$ to be n-dimensional, resulting in the form presented in our assumption. This generalization enables us to effectively capture the complex relationships within the data and account for multidimensional outcomes.
>
> **Q9: Prop 1.**
>
> In proposition 1 j simply symbolizes the vector $(g_1^T h_1, \cdots, g_n^{T} h_n)$. In proposition 1, we show that given a prognostic score, we can always find a corresponding balanced prognostic score using this formulation.
>
> **Q10: Zero overlap is not theoretically supported.**
>
> Both theoretical analysis and experiment results show our superiority on zero overlap.
> Understanding why zero-overlapping works can be seen from result 3 in theorem 1. In this case, vector of molecular properties for each molecule function as $\tilde{j_t}$in assumption 4.2, and the treatment effect is realized as the inner product between balanced prognostic score and molecular properties. As such, since the ground-truth molecular properties are given, the accuracy of the learned outcomes directly depend on the learned balanced prognostic scores. Theorem 1 result 3 tells us that if we can approximate $j_t$ arbitrarily well, then we can can in fact upper bound the outcome estimation error.
> Meanwhile, the experiments in Sec 5.4 show that our learned balanced prognostic score can even generalize to out-of-distribution treatment as well.
>
> **Q11: We can always derive a balanced prognostic score.**
>
> Same as Q8.
> Assumption 4.4 can be seen as a generalization of assumption 3 in [8], where the outcome is parameterized as $Y = g(X)^T h(T) + \epsilon$. Assumption 4.4 essentially generalizes assumption 3 in [8] from 1-dimensional to n-dimensional(We have demonstrated this in Sec4.1). In our assumption, we consider $Y$ itself to be a prognostic score as it serves as a sufficient statistic for the outcome. Additionally, $g(X)$ is a balanced prognostic score since $Y$ is conditionally independent of $X$ given $g(X)$. To extend this assumption, we allow $Y$ to be n-dimensional, resulting in the form presented in our assumption. This generalization enables us to model high-dimensional prognostic score, instead of just one-dimesnional.
>
> **Q12: Potential confusion of the symbols in the assumptions and equations.**
>
> Thanks for your comments, we will standardize all symbols in the revision to avoid potential ambiguity or reader confusion.
>
> [1]  Wu, et al. β-INTACT-VAE: IDENTIFYING AND ESTIMATING CAUSAL EFFECTS UNDER LIMITED OVERLAP. ICLR 2022.
>
> [2] Khemakhem, et al. Variational autoen- coders and nonlinear ica: A unifying framework. AISTATS 2020.
>
> [3]  Neal. Introduction to Causal Inference.
>
> [4] Wu, et al. Learning decomposed representation for counterfactual inference. TKDE 2022.
>
> [5] Cheng, et al. Learning disentangled representations for counterfactual regression via mutual information minimization SIGIR 2022.
>
> [6] Zhang, et al. Treatment effect estimation with disentangled latent factors. AAAI 2021.
>
> [7] Chen et al. Isolating sources of disentanglement in variational autoencoders. NeurIPS 2018.
>
> [8] Kaddour, et al. Causal effect inference for structured treatments. NeurIPS 2021.
>
> [9]  Hassanpour et al. Learning disentangled representations for counterfactual regression. ICLR 2020.
>
> [10] Schölkopf, et al. Toward causal representation learning. Proceedings of the IEEE 2021.

---

> ### Comment · Reviewer_DxND · 2023-11-20
>
> Thanks for the rebuttal.
>
> Indeed, I was misled to think you meant to assume the 3 hidden variables are identified up to injective mappings. I suggest writing the 1st conclusion as: "if either (line break) 1) … not empty mapping, or (line break) 2) … non-empty mapping and … (line break), then …". The “Otherwise if” was non-standard mathematical writing and misleading.
>
> But, **most of my concerns about the main theoretical claims remain**. Particularly:
>
> **Markov condition.** Surely “The conditional independence of Y and Z3 given T can be satisfied”, but your MI is not conditional on T.
>
> **Injective is unrealistic.** g takes three (at least) 1-dim variables, then the domain is (at least) 3-dim. For injective functions, dim(image) ≥ dim(domain).
>
> **Assum 4.2**. My concern was not answered.
>
> **Assum 4.4.** I cannot understand the practical and technical implications of your high-dim generalization.
>
> **Prop 1.** In math writting, if we see a “where j =” clause, then j should be mentioned before this clause.
>
> Just in case, it seems there are no revisions of the paper?

---

> ### Author Response · Authors · 2023-11-21
> **Response to Reviewer DxND**
>
> Thanks for your concerns and suggestions, we have updated the revision in which some essential clarification are added.
>
> **Markov condition:** Thanks for pointing out that, and our MI is conditional on T. It is a typo, and we have updated it in the revision. We have updated the assumption to accurately reflect the local Markov condition. This update on assumption will not impact the fundamental methodology of our proof.
>
> **Injective is unrealistic:** There is some misunderstanding in the dimension of $Z$. As mentioned earlier, Z1/Z2/Z3/Z4 are not one-dimensional, and their dimensions can be hyperparameterized. In fact, the dimension of Z4 can be larger than the sum of dimension of Z1, Z2, and Z3.
>
> **Assumption 4.2:** Thanks for your concern again. This second equality has been rigorously demonstrated in [1], and we have highlighted it in the revision to direct interested readers to the relevant source.
>
> **Assumption 4.4:** As mentioned earlier, this assumption is essentially a generalization of assumption 3 in [2], where the outcome/prognostic score is not limited to being one-dimensional. This allows us to model a high-dimensional prognostic score. By assuming that each dimension of the prognostic score can be factorized as an inner product between g(X) and h(T), we obtain our proposed form.
>
> **Proposition 1:** Thanks for your suggestion. We have made the update in our revision, removing the ``where j =" clause.
>
> We hope that our reply can address all your concerns, and we are more than willing to engage in further discussion.
>
> [1] Zhang, et al. "Treatment effect estimation with disentangled latent factors. AAAI 2021.
>
> [2] Kaddour, et al. Causal effect inference for structured treatments. NeurIPS 2021.

---

> ### Author Response · Authors · 2023-11-22
> **Response to Reviewer DxND**
>
> Dear reviewer,
> Thank you for taking the time to review our paper.
>
> It appears that most of your concerns on theory stem from a **misunderstanding of our work rather than any inherent drawbacks**. We sincerely hope that our reply can adequately address all your concerns and provide you with a clear understanding of our research.
>
> We would be more than happy to engage in further discussion with you and address any remaining questions you may have.

---

### Official Review · Reviewer_AjMV · 2023-10-24

**Soundness:** 3 good
**Presentation:** 3 good
**Contribution:** 3 good
**Rating:** 6
**Confidence:** 4

**Summary:**

The authors investigated the challenge of estimating treatment effects when there's limited overlap. They emphasized that overlap need not be present in the covariate space; instead, it suffices for overlap to exist within a latent representation. To address this, they introduced a disentangled identifiable Variational Autoencoder that effectively separates adjustment, instrumental, and confounder variables. Their experiments demonstrated that their approach outperforms other baseline methods, showcasing its superior performance.

**Strengths:**

They attempted to acquire a disentangled representation, effectively segregating confounders, instrumental variables, and adjustment variables. The results from their experiments unequivocally demonstrate a notable improvement in performance.
They used the idea from chen et al. 2018 to achieve a disentangled representation.

**Weaknesses:**

I wish I could see the results for a simple VAE without ELBO decomposition, to see how much improvement could happen. it is not clear to me how much this improvement is coming from elbo decomposition.

**Questions:**

Questions:
1. I find the decoder structure in Figure 1 unclear. It's not evident to me whether we are reconstructing observed variables from latent variables, or if we need to supply T and Y as signals to the model.
2. Equation 12's inference factorization isn't immediately clear to me. It would be greatly appreciated if the authors could provide an explanation in the appendix.
3. There are some assumptions mentioned, such as the injective nature of certain functions. Were these assumptions followed in the implementation, or were they primarily included for mathematical purposes?
4. Is it necessary to include a z4 in the model?
5. Was there any hyperparameter to balance the contribution of different losses to the final loss?
6. In section 5.3, I assumed we would observe a drop in performance in other methods while your method maintained a constant performance. However, this doesn't appear to be the case, and varying the level of limited overlap doesn't seem to affect the performance of other methods.

---

> ### Author Response · Authors · 2023-11-15
> **Response to Reviewer AjMV**
>
> **Q1: Results for a simple VAE without ELBO decomposition.**
>
> Thank you raising a really good point! We have performed ablations to verify this, where we take the optimal hyperparameters as reported in the paper and substitute it with a VAE without ELBO decomposition. We conducted our ablation study over replication 50-99 on the IHDP dataset. DIRE achieves an in-sample pehe of $0.473 \pm 0.039$ and an out-sample pehe of $0.545 \pm 0.069$, whereas the one with a simple VAE with the same architecture achieves an in-sample pehe of $0.562 \pm 0.078$ and an out-sample pehe of $0.634 \pm 0.093$. As can be seen, the ELBO decomposition trick improves the result by a large margin. As such, the ELBO decomposition trick is critical to realizing our theory.
>
> We consider the ELBO decomposition trick to be a crucial element in realizing our theory. In [2], the ELBO is decomposed into three distinct terms: the mutual information term between the latent factors and the covariates, the total correlation among the latent factors, and the divergence with the prior.
>
> The mutual information term plays a vital role in cleanly separating the information in the incoming node from the current node, such as $q(z_1 | t, z_4)$ and $q(z_3 | z_4, t)$. Since $z_1$ captures the adjustment variables, conditioning on $t$ and $z_4$ and penalizing the mutual information between $z_1$ and $t/z_4$ can be seen as a strategy to extract information from $z_4$ that is unrelated to $t$.
>
> Regarding the total correlation term, our objective is to learn representations that adhere to the independent causal mechanisms [7]. By combining the ELBO decomposition trick with our encoder/decoder structure, we perceive it as a critical and elegant approach to acquire disentangled and identifiable latent representations for the underlying data generation process. This integration allows us to effectively capture the complex relationships within the data and interpret them in a meaningful manner.
>
> **Q2: T and Y as signals to the model.**
>
> On a high level, as noted in [1], auxiliary information is needed to achieve identifiable representation, whereas unsupervised learning of identifiable representations is theoretically impossible. As such, T and Y are needed as supervision signals to obtain identifiable latent representations that give rise to $X$.
>
> **Q3: inference Factorization.**
>
> Thank you for your feedback. We will definitely take it into account for our future revisions. In a broader sense, when analyzing a node, any non-parent nodes in the decoder can be used as its parents in the encoder. By applying this process greedily, we are able to derive an appropriate encoder structure. A detailed example demonstrating this concept can be found in [3] section 2.2. Take the adjustment node in the encoder as example. Since the input node of the adjustment in the encoder is the treatment indicator and a latent variable $z_4$ that essentially represents $X$, i.e., $q(z_1 | z4, t)$, the mutual information term in the ELBO decomposition trick enable us to elicit variables in $z_4$ that are unrelated to $t$, resulting in the definition of the adjustments.
>
> **Q4: injective nature of certain functions.**
>
> Regarding the assumption of injectivity, the key premise is that the mapping from the latent factors to the covariates is injective, as demonstrated in [1] theorem 1 ii). Consequently, in the case of [4], the theory relies on the latent embedding being restricted to a dimension of 1 in order to reconstruct Y. In contrast, our approach allows for a learned representation with a dimension no higher than that of the covariates, enabling us to model a broader range of complex problems. This relaxation allows us to handle a richer class of scenarios compared to the dimension-restricted latent embedding in [4].

---

> ### Author Response · Authors · 2023-11-15
> **Response to Reviewer AjMV**
>
> **Q5:  z4 in the model.**
>
> Theoretically we can get rid of z4 and everything still works. Empirically we find that including z4 gives us better estimation. Such result has been explored in [6] as well.
>
> **Q6: hyperparameter to balance the contribution of different losses to the final loss.**
>
> To optimize each component, we employ hyperparameterization and perform a thorough search over a validation dataset. In order to streamline the hyperparameter search process, we incorporate [8] into our training workflow, utilizing [9]. We ensure a fair comparison by conducting an equal number of searches for each model. This approach allows us to effectively explore the hyperparameter space and find optimal configurations for each model. We have detailed our hyperparameters search range in our appendix.
>
> **Q7: More explanation on Fig 2.**
>
> While the figure may appear small and difficult to discern, as depicted in fig. 2, our proposed method maintains robust performance as the degree of limited overlapping increases. In contrast, both [5] and [4] demonstrate decreasing performance as omega increases from 5.0 to 15.0. The error bars of [5] and [4] are both rising, corresponding to a larger PEHE, whereas our method's error bar even decreases with increasing omega.
>
> Specifically, when omega = 10.0, DR-CFR's out-sample pehe value is $3.65 \pm 0.15$, whereas DIRE's pehe value is $3.33 \pm 0.12$. When omega = 15.0, DR-CFR's out-sample pehe value is $3.74 \pm 0.15$, whereas DIRE's pehe value is $3.30 \pm 0.13$. DR-CFR's pehe error is increasing while the degree of limited overlap increases, whereas DIRE's performance remains robust when confronted with extreme limited overlap.
>
> [1] Khemakhem, et al. Variational autoen- coders and nonlinear ica: A unifying framework. AISTATS 2020.
>
>
> [2] Chen, et al. Isolating sources of disentanglement in variational autoencoders. NeurIPS 2018.
>
> [3] Louizos, et al. "The variational fair autoencoder." ICLR 2016.
>
> [4] Wu, et al. Identifying and estimating causal effects under weak overlap by generative prognostic model. ICLR 2022.
>
> [5] Hassanpour, et al. Learning disentangled representations for counterfactual regression. ICLR 2020
>
> [6] Kingma, et al. "Semi-supervised learning with deep generative models." NeurIPS (2014).
>
> [7] Schölkopf, et al. Toward causal representation learning. Proceedings of the IEEE 2021.
>
> [8] Liam Li, et al. A system for massively parallel hyperparameter
> tuning. MLSys 2020.
>
> [9] Philipp Moritz, et al. Ray: A distributed framework for emerging {AI} applications. OSDI 2018.

---

### Official Review · Reviewer_N81L · 2023-10-28

**Soundness:** 3 good
**Presentation:** 3 good
**Contribution:** 3 good
**Rating:** 6
**Confidence:** 2

**Summary:**

This paper studies the identifiability of treatment effects under limited overlap, but with latent adjustments, confounders, and instruments. Under a general causal graph model, the authors show that overlapping conditions can be sustantially relaxed, and treatment effects can extend to non-overlapping regions. Experiments also show that the proposed method achieves superior performance compared with competing methods in various benchmarks.

**Strengths:**

1. Significance and contribution.

Treatment effect estimation beyond overlap is an important problem. This paper contributes to this literature by proposing a model that enables treatment effect generalization and methods to achieve so.

2. Quality and clarity.

This paper is clearly written, with discussions from time to time that address possible confusions. The experiments are thorough and provide concrete support to the technical part.

**Weaknesses:**

Discussion on the model

The identifiability of treatment effects relies crucially on the model. While some part such as outcome DGP is discussed so that readers understand they are weaker than existing literature, assumption 4.3 and 4.4 for treatment and prognostic score may need more justification.

**Questions:**

1. I don't really get why a bPGS can always be derived based on a PGS (right after Proposition 1). Can you provide more discussion?

2. Besides justifying the model assumptions, is there a way to verify this model is reasonable given a dataset?

---

> ### Author Response · Authors · 2023-11-15
> **Response to Reviewer N81L**
>
> **Q1: Needs more discussion. Identifiability crucially relies on model.**
>
> Thank you for your feedback! We will add more discussions in the revision.
>
> A crucial ingredient in realizing identifiability is the ELBO decomposition trick. In [2], they decompose the ELBO into three components: the mutual information term between the latent factors and the covariates, the total correlation among the latent factors, and the divergence with the prior. The mutual information term plays a vital role in cleanly separating the information in the incoming node from the current node, such as $q(z_1 | t, z_4)$ and $q(z_3 | z_4, t)$. Since $z_1$ captures the adjustment variables, conditioning on $t$ and $z_4$ and penalizing the mutual information between $z_1$ and $t/z_4$ can be seen as a strategy to extract information from $z_4$ that is unrelated to $t$.
>
> The total correlation term is designed to learn representations that adhere to independent causal mechanisms [7]. When combined with our encoder/decoder structure, the ELBO decomposition trick provides a critical and elegant approach to learning disentangled and identifiable latent representations for the underlying data generation process. It allows us to extract meaningful information and capture the causal relationships within the data.
>
> Although not explicitly stated, various disentanglement-based methods for treatment effect estimation, such as [3-6], utilize prognostic score based and balancing score based losses. Recognizing their effectiveness, we also incorporate these losses into our framework to facilitate the learning of latent embeddings. By integrating these losses into our approach, we aim to enhance the disentanglement and interpretability of the learned representations, ultimately improving the accuracy and robustness of our treatment effect estimation.
>
> **Q2: assumption 4.3, assumption 4.4.**
>
> Thanks for you comments, and we will provide more details of these assumption in the revision.
> Assumption 4.3 serves as a formalization that the balancing score is a sufficient statistic for $T$ and is included for completeness. On the other hand, assumption 4.4 can be seen as a generalization of assumption 3 in [1]. In [1], the outcome is parameterized as $Y = g(X)^T h(T) + \epsilon$. In our assumption, we consider $Y$ itself to be a prognostic score as it serves as a sufficient statistic for the outcome. Additionally, $g(X)$ is a balanced prognostic score since $Y$ is conditionally independent of $X$ given $g(X)$. To extend this assumption, we allow $Y$ to be n-dimensional, resulting in the form presented in our assumption. This generalization enables us to model high-dimensional prognostic score, instead of just one-dimesnional ones.
>
> **Q3: bPGS based on PGS.**
>
> Consider the special case where we take PGS as Y, and $Y = g(X)^T h(T) + \epsilon$ as mentioned above. We then have $Y$ as the PGS and $g(X)$ as the bPGS. we essentially generalize this special case to n-dimensional PGS to model high-dimensional outcome. Moreover, since the decoder is injective (an assumption made for [8], assumption 4.1 in our case), our learned bPGS can be interpreted as the inverse mapping in the decoder, which is derived solely from $X$.
>
> **Q4: verifying model is reasonable given a dataset.**
>
> We have performed ablations to verify this, where we take the optimal hyperparameters as reported in the paper and substitute it with a VAE without ELBO decomposition. We conducted our ablation study over replication 50-99 on the IHDP dataset. DIRE achieves an in-sample pehe of $0.473 \pm 0.039$ and an out-sample pehe of $0.545 \pm 0.069$, whereas the one with a simple VAE with the same architecture achieves an in-sample pehe of $0.562 \pm 0.078$ and an out-sample pehe of $0.634 \pm 0.093$. As can be seen, the ELBO decomposition trick improves the result by a large margin. As such, the ELBO decomposition trick is critical to realizing our theory.
>
> [1] Kaddour, et al. Causal effect inference for structured treatments. NeurIPS 2021.
>
> [2] Chen, et al. Isolating sources of disentanglement in variational autoencoders. NeurIPS 2018.
>
> [3] Hassanpour, et al. Learning disentangled representations for counterfactual
> regression. ICLR 2020.
>
> [4] Cheng, et al. Learning disentangled representations for counterfactual regression via mutual information minimization. SIGIR 2022.
>
> [5] Zhang, et al. "Treatment effect estimation with disentangled latent factors. AAAI 2021.
>
> [6] Wu, et al. Learning decomposed representation for counterfactual inference. TKDE 2022.
>
> [7] Schölkopf, et al. Toward causal representation learning. Proceedings of the IEEE 2021.
>
> [8] Khemakhem, et al. Variational autoen- coders and nonlinear ica: A unifying framework. AISTATS 2020.

---

> > ### Comment · Reviewer_N81L · 2023-11-20
> >
> > Thank you for your discussion! It's helpful to add such results in the revision, and I'll keep my score.

---

### Official Review · Reviewer_QXYk · 2023-11-04

**Soundness:** 3 good
**Presentation:** 3 good
**Contribution:** 3 good
**Rating:** 6
**Confidence:** 3

**Summary:**

In the paper, the authors tackle an important problem in causal inference: estimating individual-level treatment effects when there is limited overlap in covariates across treatment groups. To be specific, traditional causal inference methods require substantial overlap in covariates between different treatment groups to accurately estimate treatment effects. The paper focuses on cases where this condition is not met, which is challenging for existing methods. The authors propose a solution that allows for the estimation of treatment effects when covariate overlap is insufficient. They achieve this by recovering two types of scores:
   - **Balanced Prognostic Score**: Reflects the expected outcome of an individual without treatment.
   - **Balancing Score**: Indicates the probability of an individual receiving a particular treatment, given their covariates.

The Disentangled Identifiable vaRiational autoEncoder (DIRE) is introduced as a key technical tool. It is a model that disentangles the factors of variation in the data while maintaining identifiable features. Besides, the paper presents theoretical arguments for how the balanced prognostic score effectively manages the issue of limited overlap, and how it can adapt to scenarios where there is zero overlap, addressing out-of-distribution treatments.

Finally, the authors conduct extensive experiments that benchmark their method against others, especially in scenarios with binary treatments and in complex situations where traditional methods may fail due to limited covariate overlap.

**Strengths:**

1. The paper tackles the critical issue of non-overlap in causal inference, a problem that, if unaddressed, renders many causal analyses ineffective. By confronting this problem head-on, the research addresses a fundamental bottleneck in causal methodology, ensuring that the insights drawn from such analyses are both valid and applicable in more realistic scenarios where perfect overlap is not present.

2. The authors have conducted an extensive array of simulation studies to showcase the performance of their proposed method. These simulations are critical for demonstrating the method's effectiveness across a variety of conditions and benchmarks.

3. The paper excels in its articulate presentation. It defines the problem of estimating individual-level treatment effects in scenarios with non-overlapping covariates succinctly. The proposed method, including the innovative use of the Disentangled Identifiable vaRiational autoEncoder (DIRE), is described with a clarity that ensures readers are able to grasp both the significance and the application of the research.

4. The theoretical underpinnings of the paper are robust and effectively illuminate the concepts behind the methodology. The theoretical sections can support the practical aspects of the proposed method but also enhance the reader's comprehension of why the method works.

**Weaknesses:**

Please see more details in the following Questions parts.

**Questions:**

1. On page 3, the author elucidates the concepts of non-overlapping and limited overlapping with clarity. Yet, in the experimental analysis, specifically in Q2, when introducing a 'degree' of non-overlapping, the definition remains ambiguous. It is crucial for the reader to understand the extent to which this method can effectively operate within various levels of non-overlap. Could the author provide a more detailed explanation? Additionally, the experiment study (Q2) suggests that the proposed "DIRE" method's performance is unaffected by the degree of non-overlapping. This assertion underscores the robustness of the method, but it warrants a deeper explanation to substantiate such a claim.

2. Section 4.3 discusses the integration of an 'ELBO decomposition trick' into the method, which contributes to the final loss function. The specific advantages of incorporating this technique, particularly in the context of addressing limited overlap issues, have not been fully articulated. What incremental value does this approach provide, and how does it interact with the other components of the loss function, namely the prognostic score-based and balancing score-based losses? If the loss function were simplified to include only these two components, how might that impact the method's performance?

3. In section 5.4, the paper navigates the complex terrain of structured treatment settings. An elucidation of the inherent challenges within such settings would greatly benefit the reader. Does this term imply a scenario with a multitude of treatments amongst which certain structural patterns are discernible? If so, to the best of my knowledge, two interesting studies can be noted: one from the Journal of Machine Learning Research (JMLR) in 2023: "Learning Optimal Group-structured Individualized Treatment Rules with Many Treatments", and another from the Neural Information Processing Systems (NIPS) conference in 2022: "Learning Individualized Treatment Rules with Many Treatments: A Supervised Clustering Approach Using Adaptive Fusion". Both papers address situations of limited overlap amid an array of many treatments, focusing primarily on the refinement of individualized treatment rules. However, there appears to be a difference in their approaches compared to the one presented in this paper. Could the author expound on the distinctions and potential synergies between these methodologies and the current approach under discussion?

---

> ### Author Response · Authors · 2023-11-15
> **Response to Reviewer QXYk**
>
> **Q1: Explanation on limited overlapping performance.**
>
> Thank you for pointing out that we need more explanations! Similar to [1], we quantify the degree of limited overlap by examining the percentage of observed values that have a probability less than 0.001 for one of $p(t | x)$. When oemga = 15.0, 80 percent of the observed values are limited overlapping, making our dataset more difficult than that of [1] in terms of limited overlapping. We wll include plot similar to the one in Appendix E.1 in [1] in the revision.
>
> In fig. 2, our proposed method maintains robust performance as the degree of limited overlapping increases. In contrast, both [5] and [4] demonstrate decreasing performance as omega increases from 5.0 to 15.0. The error bars of [5] and [4] are both rising, corresponding to a larger PEHE, whereas our method's error bar even decreases with increasing omega.
>
> Specifically, when omega = 10.0, DR-CFR's out-sample pehe value is $3.65 \pm 0.15$, whereas DIRE's pehe value is $3.33 \pm 0.12$. When omega = 15.0, DR-CFR's out-sample pehe value is $3.74 \pm 0.15$, whereas DIRE's pehe value is $3.30 \pm 0.13$. DR-CFR's pehe error is increasing while the degree of limited overlap increases, whereas DIRE's performance remains robust when confronted with extreme limited overlap.
>
> Intuitively, if we have successfully learned a balanced prognostic score (bPGS), the limited overlapping regions within a treatment arm $j_t$ can be inferred, provided that $j_t$ converges closely to the ground truth. This corresponds to result 3 in theorem 1.
>
> **Q2: benefit of integrating the 'ELBO decomposition trick' into the method, the prognostic score based and balancing score based losses.**
>
> Thanks for your attention. We have performed ablations to verify this, where we take the optimal hyperparameters as reported in the paper and substitute it with a VAE without ELBO decomposition. We conducted our ablation study over replication 50-99 on the IHDP dataset. DIRE achieves an in-sample pehe of $0.473 \pm 0.039$ and an out-sample pehe of $0.545 \pm 0.069$, whereas the one with a simple VAE with the same architecture achieves an in-sample pehe of $0.562 \pm 0.078$ and an out-sample pehe of $0.634 \pm 0.093$. As can be seen, the ELBO decomposition trick improves the result by a large margin. As such, the ELBO decomposition trick is critical to realizing our theory.
>
>
> We consider the ELBO decomposition trick as a crucial tool for realizing our theory. In [2], they decompose the ELBO into three components: the mutual information term between the latent factors and the covariates, the total correlation among the latent factors, and the divergence with the prior. The mutual information term plays a key role in separating the information in the incoming node from the current node, such as $q(z_1 | t, z_4)$ and $q(z_3 | z_4, t)$. By conditioning on $t$ and penalizing the mutual information between $z_1$ and $t/z_4$, we aim to capture information in $z_4$ that is unrelated to $t$, while $z_1$ models the adjustment variables.
>
> The total correlation term is designed to learn representations that adhere to independent causal mechanisms [7]. When combined with our encoder/decoder structure, the ELBO decomposition trick provides a critical and elegant approach to learning disentangled and identifiable latent representations for the underlying data generation process. It allows us to extract meaningful information and capture the causal relationships within the data.
>
> Although not explicitly stated, various disentanglement-based methods for treatment effect estimation, such as [3-5, 8], utilize prognostic score based and balancing score based losses. Recognizing their effectiveness, we also incorporate these losses into our framework to facilitate the learning of latent embeddings. By integrating these losses into our approach, we aim to enhance the disentanglement and interpretability of the learned representations, ultimately improving the accuracy and robustness of our treatment effect estimation.

---

> ### Author Response · Authors · 2023-11-15
> **Response to Reviewer QXYk**
>
> **Q3: Distinctions and potential synergies between [6, 9] and the current approach under discussion.**
>
> Thank you for bringing up these interesting papers! We will certainly discuss distinctions and relationship with these works in the revisions.  In a similar vein to [6], we represent each treatment as a vector of treatment-specific parameters. In both our baseline method, SIN-with-Aux-Info, and our main method, DIRE, we utilize a fixed treatment representation consisting of molecular properties. This choice allows us to demonstrate that we learn a more effective balanced prognostic score compared to SIN-with-Aux-Info. There is potential for synergy between our method and [6] as follows:
>
> i) Incorporating the idea of clustering similar treatments from [6] could further enhance our approach by grouping treatments together, particularly in scenarios where there are numerous similar treatments. This clustering approach could facilitate the identification of the balanced prognostic score.
>
> ii) In [6], variable selection methods are primarily employed to identify homogeneous and heterogeneous variables. Our method could be potentially coupled with this approach to enhance the identification of homogeneous/heterogeneous variables and improve the accuracy of treatment effect estimation.
>
> By combining our method and the ideas presented in [6]., we can potentially enhance the performance and interpretability of treatment effect estimation.
>
> [1] Wu, et al. Identifying and estimating causal effects under weak overlap by generative prognostic model. ICLR 2022.
>
> [2] Chen, et al. Isolating sources of disentanglement in variational autoencoders. NeurIPS 2018.
>
> [3] Hassanpour, et al. Learning disentangled representations for counterfactual
> regression. ICLR 2020.
>
> [4] Cheng, et al. Learning disentangled representations for counterfactual regression via mutual information minimization. SIGIR 2022.
>
> [5] Zhang, et al. "Treatment effect estimation with disentangled latent factors. AAAI 2021.
>
> [6] Ma, et al. Learning Individualized Treatment Rules with Many Treatments: A Supervised Clustering Approach Using Adaptive Fusion. NeurIPS 2022.
>
> [7] Schölkopf, et al. Toward causal representation learning. Proceedings of the IEEE 2021.
>
> [8] Wu, et al. Learning decomposed representation for counterfactual inference. TKDE 2022.
>
> [9] Ma, et al. Learning Optimal Group-structured Individualized Treatment Rules with Many Treatments. JMLR 2023.

---

### Meta-Review · Area_Chair_DMan · 2023-12-06

**Metareview:**

Dealing with weak overlap is a vital practical point, and the contributions proposes advancements by taking the problem into the latent space. There is much to appreciate in the manuscript in this regard.

There was a productive discussion on the possible shortcomings of the method. Some clarifications may have been underappreciated in the discussion - for instance, I was able to understand what the authors meant about being able to choose the dimensionality of $Z_4$ (it still doesn't mean that the conditions are easy or intuitive to verify...)

Some of the other rebuttal points are less clear. It was simply stated that lack of conditioning on $T$ on the definition of the MI score is a typo with no consequences, but this is not considerate of the readers. Details of *how* it has no consequences is something that authors should provide in a rebuttal. It is not a job of the ICLR reviewers to re-read a paper multiple times when it had issues at submission time.

There are other points which also not very clear. We agree that the model in Figure 1 is not nonparametrically identifiable (as there are latent backdoors) and so there is a need to convince the reader what other technical assumptions will be able to recover it. Those always require careful exposition and sometimes I didn't feel too confident that the writing was delivering it. For instance, I tried hard to understand how the second equality in Eq. (3) "follows" from the "do-calculus ... in Fig 1", or how it is "rigorously demonstrated" from Zhang et al. (2021), or from Wu and Fukumizu (2021) - partially because I don't know whether this is a definition of $e_2$ or whether there was something else that implied additive error structure.  My impression is that the second equality is just an assumption, not something that follows from deeper principles, but arguments in the discussion and paper threw me off.

All in all, I hold the opinion that the paper gets many things right, but there are a handful of less clear passages that cumulatively takes the toll on readers trying to understand the finer details.

**Justification For Why Not Higher Score:**

There is many positive points to be said about the paper, but some of the finer details remove some of the confidence in them. The main one I would point out is the mismatch in the definition of mutual information and what the model assumptions entail. It is not clear what the theoretical and empirical implications it has, and the rebuttal doesn't quite deliver the confidence I would like to have gotten from it.

**Justification For Why Not Lower Score:**

N/A

---

### Decision · Program_Chairs · 2024-01-16

Reject